# High-Order Dynamics Modeling of Time Series with Attractor-Guided Adaptive Filtering

## Abstract

Explicit, equation-discovery models promise transparent mechanisms and strong extrapolation for time-series dynamics. Yet most existing methods impose first-order structure, even when the true system depends on multiple lags. This mismatch is typically absorbed by inflating the latent state via ad-hoc augmentation, which erodes identifiability, complicates learning, and weakens interpretability. Compounding the issue, defaulting to Kalman-style updates in nonlinear or weakly stable regimes is brittle: inference degrades away from fixed points, biasing parameter estimates and reducing predictive reliability.

We introduce a framework for *adaptive high-order dynamics modeling*. Given an $m$-dimensional series, we *initialize the latent dimension to $m$* and estimate the Markov order $p$—the minimal number of past states needed to predict the next—via a conditional mutual information test. Rolling statistics assess proximity to attractors and drive *stability-aware* filter selection. Starting from $(p, m)$, an inference–learning loop evaluates candidate structures and guides a unidirectional search that converges to $(\hat{p}, \hat{m})$ together with the associated system parameters. Across benchmark datasets, the resulting models yield more flexible latent dynamics and consistently improve predictive accuracy over state-of-the-art baselines.

## 1 Introduction

Time–series analysis benefits most when models make the governing mechanisms explicit rather than merely fitting trajectories. We therefore focus on *explicit dynamical equation modeling*: learning closed-form latent transition rules and observation maps that support fixed-point and stability analysis, controllability, and principled intervention design (e.g., Kalman (1963); Zarchan (2005)). In contrast to black-box sequence models that excel at prediction but offer limited mechanistic insight (e.g., Ismail Fawaz et al. (2019); Baier et al. (2023)), explicit equations enable extrapolation under structural priors and clear separation of process and measurement noise.

Two research lines are especially relevant. First, equation-discovery methods such as SINDy and its variants recover parsimonious nonlinear dynamics from data by sparse regression over libraries of candidate terms Brunton et al. (2016); Champion et al. (2020); Kaptanoglu et al. (2022); Boninsegna et al. (2018); Bertsimas & Gurnee (2023). Symbolic regression broadens the search space beyond fixed libraries to identify tractable analytical formulas La Cava et al. (2018); Burlacu et al. (2020); Landajuela et al. (2022); Udrescu & Tegmark (2020); Shojaee et al. (2023). These approaches provide readable models when states (or their derivatives) are directly observed, but they neither infer latent trajectories nor handle partial observability gracefully; moreover, reliance on numerical differentiation can be brittle under noise Mangan et al. (2017); Grünwald (2007).

Second, state-space modeling couples transition and observation equations and performs latent-state inference via filtering/smoothing Akaike (1974); Pearl (1982); Ghahramani & Roweis (1998); Fox et al. (2008); Chen & Poor (2022); Liu & Hauskrecht (2015). While this line affords noise robustness and missing-data handling, much of it either enforces linear transitions or—when nonlinear—retains a *first-order* Markov assumption, pushing higher-order memory into inflated latent dimensions that erode interpretability Foster et al. (2020); Kowshik et al. (2021); Sattar & Oymak (2022); Kakade et al. (2011).

Beyond these two lines, a substantial body of system-identification work reconstructs dynamics through Hankel embeddings and delay-coordinate methods. Classical approaches in nonlinear time-

series analysis lift the data into a Hankel (trajectory) matrix, enabling linear or bilinear operators to approximate the underlying flow Abarbanel (1996); H. Tu et al. (2014). Empirical Dynamic Modeling (EDM) Sugihara & May (1990); Sugihara et al. (2012) further exploits delay-coordinate reconstructions to perform state-space prediction without prescribing an explicit parametric form. More recently, DeepEDM Ghosh et al. (2025) integrates neural approximators into the EDM pipeline via nonlinear manifold learning. Although powerful, these Hankel/EDM approaches operate in reconstructed observable spaces rather than latent dynamical coordinates, and typically do not yield explicit, closed-form transition equations. They therefore complement but do not replace explicit latent-dynamics modeling.

Among explicit latent-dynamics methods, **LaNoLeM** Fujiwara et al. (2025) is notable for recovering closed-form nonlinear transitions within a latent state-space. However, it still presumes first-order dynamics and primarily relies on Kalman-style updates, which are well-behaved near fixed points but degrade in strongly nonlinear or weakly stable regimes.

We propose a unified framework for *adaptive high-order state–space modeling* that explicitly accommodates multi-step temporal dependencies and introduces *stability-aware* inference. Given an $m$-dimensional series, we initialize the latent dimension to $m$ and obtain a preliminary Markov order $p_0$ via a conditional mutual information test (the Markov order is the smallest number of past states sufficient for next-step prediction). We then compute rolling-window statistics to quantify proximity to attractors; this stability proxy adaptively selects particle filtering in unstable regions and Kalman filtering near attractors. Starting from $(p_0, m_0)$, a structured unidirectional search evaluates each candidate via an inner *inference–learning loop* that jointly estimates latent trajectories and system parameters. The procedure converges to an optimal pair $(\hat{p}, \hat{m})$ together with an explicit model of the dynamics. Figure 1 provides an overview.

Our contributions are threefold:

- A **stability-aware filtering principle** that chooses between Kalman and particle filters based on proximity to attractors, improving robustness in unstable regimes while retaining efficiency near equilibria.
- A **structured search strategy** that jointly identifies the Markov order $\hat{p}$ and latent dimension $\hat{m}$ via a single-direction walk guided by the inference–learning loop, avoiding combinatorial explosion.
- A **complete recovery framework** for explicit dynamical systems, integrating temporal-dependence estimation, stability-guided inference, and parameter learning to improve predictive accuracy and interpretability across diverse benchmarks.

## 2 PRELIMINARIES

### 2.1 MARKOV ORDER

Temporal dependence means that future evolution is shaped by past history. We formalize this with a state–transition function $f$ on latent states $\mathbf{s}_t \in \mathbb{R}^m$, which maps a segment of the past trajectory to the next state.

The simplest case is first-order dynamics, where only the most recent state matters:

$$\mathbf{s}_{t+1} = f(\mathbf{s}_t). \tag{1}$$

In many systems, however, a single lag is insufficient to capture delayed effects or accumulated interactions. We therefore allow dependence on multiple past states:

$$\mathbf{s}_{t+1} = f(\mathbf{s}_t, \mathbf{s}_{t-1}, \ldots, \mathbf{s}_{t-p+1}). \tag{2}$$

The *Markov order* $p$ is defined as the *smallest* number of lags for which such a representation holds—no shorter history suffices. Intuitively, $p$ characterizes the system's minimal memory length: the effective horizon over which past states influence $\mathbf{s}_{t+1}$.

### 2.2 ATTRACTORS

A fundamental concept in discrete-time dynamical systems is the *attractor*: a region of state space toward which trajectories converge under repeated iteration. Typical examples include stable fixed

points and stable periodic orbits. For clarity, we analyze the stable fixed point case as an illustrative example.

Formally, a state $\mathbf{s}^*$ is a *fixed point* of the transition map $f$ if

$$f(\mathbf{s}^*) = \mathbf{s}^*. \tag{3}$$

Consider deviations $\delta_t = \mathbf{s}_t - \mathbf{s}^*$ near $\mathbf{s}^*$. Linearizing $f$ around $\mathbf{s}^*$ yields

$$\delta_{t+1} \approx A\,\delta_t, \qquad A = Df(\mathbf{s}^*), \tag{4}$$

where $Df(\mathbf{s}^*)$ is the Jacobian matrix of $f$ at $\mathbf{s}^*$. The fixed point is (locally) stable if the spectral radius $\rho(A) < 1$, in which case perturbations decay geometrically:

$$\delta_t \approx A^t \delta_0 \;\to\; 0, \quad t \to \infty. \tag{5}$$

To make the effect of noise explicit, augment the linearization with an additive disturbance $\mathbf{w}_t \sim \mathcal{N}(\mathbf{0}, \Sigma_w)$:

$$\delta_{t+1} \approx A\,\delta_t + \mathbf{w}_t. \tag{6}$$

Let $Q_t = \mathrm{Cov}(\delta_t)$. The deviation covariance evolves under the discrete Lyapunov recursion

$$Q_{t+1} = AQ_tA^\top + \Sigma_w. \tag{7}$$

If $\rho(A) < 1$, there exists a unique positive semidefinite steady-state covariance $\Sigma_\star$ solving

$$Q_\star = AQ_\star A^\top + \Sigma_w \quad \Longleftrightarrow \quad Q_\star = \sum_{k=0}^{\infty} A^k \Sigma_w (A^\top)^k. \tag{8}$$

Thus, the impact of noise remains bounded and is attenuated near the attractor—a phenomenon we refer to as *noise compression*. An analogous analysis applies to stable periodic orbits and is deferred to Appendix A.

These notions have direct implications for inference. *Near attractors*, deviations remain bounded and linearization is accurate, so Kalman-type filtering is effective. *Far from attractors*, nonlinearities dominate; disturbances accumulate and amplify, necessitating particle-based inference.

## 3 PROPOSED FRAMEWORK

### 3.1 PROBLEM FORMULATION

We aim to recover a latent nonlinear dynamical system from an observed time series. This entails specifying (i) a *state–transition model* governing the latent dynamics and (ii) an *observation model* linking latent states to measured signals. Let $\mathbf{s}_t \in \mathbb{R}^m$ denote the latent state and $\mathbf{y}_t \in \mathbb{R}^n$ the corresponding observation. We now detail both components.

**State transition.** To capture higher–order temporal dependencies, we augment the state with $p$ lags:

$$\mathbf{x}_t = \left[\, \mathbf{s}_t^\top,\ \mathbf{s}_{t-1}^\top,\ \ldots,\ \mathbf{s}_{t-p+1}^\top \,\right]^\top \in \mathbb{R}^{pm}. \tag{9}$$

Given $\mathbf{x}_t$, the latent dynamics are modeled by a degree-$d$ polynomial expansion with Gaussian process noise:

$$\mathbf{s}_{t+1} = \mathbf{b} + \sum_{k=1}^{d} A^{(k)}\,\phi_k(\mathbf{x}_t) + \mathbf{w}_t, \qquad \mathbf{w}_t \sim \mathcal{N}(\mathbf{0}, \Sigma_w), \tag{10}$$

where $\mathbf{b} \in \mathbb{R}^m$ is a bias, $A^{(k)} \in \mathbb{R}^{m \times \binom{pm+k-1}{k}}$ are coefficient matrices, and $\phi_k(\mathbf{x}_t)$ collects all *unique* degree-$k$ monomials of $\mathbf{x}_t$. For illustration, with $\mathbf{z} = [x_0, y_0]^\top$,

$$\phi_2(\mathbf{z}) \;=\; \left[\, x_0^2,\ x_0 y_0,\ y_0^2 \,\right]^\top, \tag{11}$$

where duplicate terms such as $yx$ are omitted by construction.

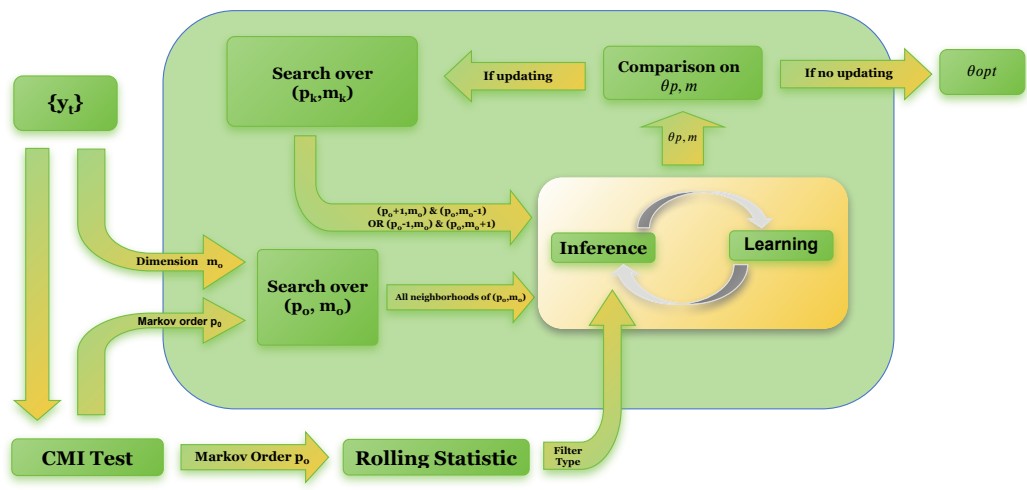

Figure 1: Framework of the proposed method.

**Observation model.** Measurements are generated by a linear map with offset and Gaussian noise:

$$\mathbf{y}_t = C\,\mathbf{s}_t + \mathbf{d} + \mathbf{v}_t, \qquad \mathbf{v}_t \sim \mathcal{N}(\mathbf{0}, \Sigma_v), \tag{12}$$

where $C \in \mathbb{R}^{n \times m}$, $\mathbf{d} \in \mathbb{R}^n$, and $\Sigma_v \in \mathbb{R}^{n \times n}$. This formulation ensures a transparent measurement channel while making identifiability explicit.

**Learning objective.** Our task is to estimate the full parameter set

$$\Theta = \{\, p,\, m,\, C,\, \mathbf{b},\, \mathbf{d},\, \{A^{(k)}\}_{k=1}^{d} \,\}, \tag{13}$$

thereby recovering both the latent order $(p, m)$ and an explicit polynomial representation of the nonlinear dynamics.

### 3.2 Initialization of Markov Order $p_0$ and State Dimension $m_0$

At the outset, we require *preliminary* values $(p_0, m_0)$ to initialize the first round of inference and learning, $p_0$ also sets the rolling-window width for stability diagnostics. Since no parametric model has been identified at this stage, these values must be chosen using nonparametric, model–free diagnostics computed directly from the observed data.

**Initialization of Markov Order $p_0$.** To quantify lagged dependence, we use *conditional mutual information* (CMI) (Cover & Thomas, 2006), which tests whether an older lag contributes predictive information beyond more recent lags. For a candidate lag $p \geq 1$,

$$I\big(y_t; y_{t-p} \,\big|\, y_{t-1}, \ldots, y_{t-p+1}\big), \tag{14}$$

which vanishes exactly when $y_{t-p}$ carries no additional information about $y_t$ given the intervening history. This motivates the population-level characterization

$$I\big(y_t; y_{t-p} \,\big|\, y_{t-1}, \ldots, y_{t-p+1}\big) = 0, \tag{15}$$

with the *true* Markov order identified as the largest $p$ satisfying equation 15.

In practice, empirical CMIs are rarely zero due to sampling noise (Kraskov et al., 2004; Frenzel & Pompe, 2007). To separate signal from noise in a distribution–free manner, we combine CMI with a permutation test (Good, 2005; Theiler et al., 1992; Schreiber & Schmitz, 2000): randomly permute

$y_{t-p}$ across time to break temporal dependence while preserving its marginal, recompute CMI on each surrogate, and compare against the observed value:

$$q_p = \frac{1}{B} \sum_{b=1}^{B} \mathbf{1}\Big\{ I^{(b)}(y_t; y_{t-p} \mid \cdot) \geq I(y_t; y_{t-p} \mid \cdot) \Big\}, \tag{16}$$

where $I^{(b)}(\cdot)$ denotes the CMI on the $b$-th permuted series, $B$ is the number of permutations, and $\mathbf{1}\{\cdot\}$ is the indicator function. A lag $p$ is declared *significant* if $q_p < \alpha$ (e.g., $\alpha = 0.05$). The initialization is then defined as

$$p_0 = \max\{ p : q_p < \alpha \}, \tag{17}$$

i.e., the longest lag whose incremental information survives rigorous null comparison—an interpretable proxy for the effective memory length of the data.

Definitions of mutual information, our CMI estimator, and the associated significance tests are deferred to Appendix B.

**Initialization of State Dimension $m_0$.** In general, the Markov order inferred at the observation layer need not equal the true latent order; they coincide only when the observation operator is invertible (Kailath, 1980; Chen, 1999; Ljung, 1999). For the linear observation model $\mathbf{y}_t = C\mathbf{s}_t + \mathbf{d} + \mathbf{v}_t$, a necessary (though not sufficient) condition for invertibility is that $C$ be square (i.e., $m = n$). Absent stronger structural assumptions, we therefore initialize the latent dimension to match the observation dimension,

$$m_0 = n, \tag{18}$$

recognizing that this is a coarse starting point used solely to seed the subsequent $(p, m)$ search.

### 3.3 FILTER SELECTION BASED ON STABILITY PROXIMITY

In the inference stage, the choice of filtering method is crucial for reliable state estimation. Our principle is to select the filter adaptively according to the trajectory's proximity to an attractor of the underlying dynamical system. Intuitively, when the system is close to a stable equilibrium, both the mean and variance of fluctuations contract; conversely, far from attractors, nonlinear propagation amplifies deviations. This motivates the use of rolling statistics as data–driven proxies for stability proximity.

Let $\{y_t\}_{t=1}^{T} \subset \mathbb{R}^d$ denote the observed $d$-dimensional time series of length $T$. Fix a window size $W$, producing $n = T - W + 1$ overlapping windows. For each window $[t, t + W - 1]$, compute the rolling mean $\mu_t \in \mathbb{R}^d$ and unbiased covariance $C_t \in \mathbb{R}^{d \times d}$:

$$\mu_t = \frac{1}{W} \sum_{i=t}^{t+W-1} y_i, \qquad C_t = \frac{1}{W-1} \sum_{i=t}^{t+W-1} (y_i - \mu_t)(y_i - \mu_t)^\top, \qquad t = 1, \ldots, n. \tag{19}$$

To normalize these statistics across time and dimensions, we compute a baseline from the earliest segment of the trajectory:

$$L_0 = \max\{10, \lfloor \sqrt{T} \rfloor\}, \qquad \mu_0 = \frac{1}{L_0} \sum_{i=1}^{L_0} y_i, \qquad S_0 = \mathrm{Cov}(y_{1:L_0}) + \epsilon I_d, \tag{20}$$

where $I_d$ is the $d \times d$ identity matrix and $\epsilon > 0$ ensures positive definiteness. The window length $L_0$ balances variance and locality: it is long enough to yield a stable covariance estimate, yet short enough to reflect a single dominant dynamical regime. Consequently, $(\mu_0, S_0)$ serves as a practical approximation of the quasi-stationary statistics within an attraction basin.

We then compress $(\mu_t, C_t)$ into two scalar proxies. The first proxy measures *mean drift* using the squared Mahalanobis distance (Mahalanobis, 1936) relative to the baseline:

$$m_t = (\mu_t - \mu_0)^\top S_0^{-1} (\mu_t - \mu_0). \tag{21}$$

This statistic is scale-invariant and reflects how far the rolling-window mean deviates from the baseline. Near a stable equilibrium $\mathbf{s}^*$, we may linearize the dynamics as $\delta_{t+1} = A\delta_t + w_t$ with

$\delta_t = s_t - s^*$. When $\rho(A) < 1$, $\delta_t$ converges in mean to zero under zero-mean disturbances. If the baseline window lies in the same attraction basin so that $\mu_0 \approx \mu^*$, then $(\mu_t - \mu_0) \to 0$ and the Mahalanobis drift $m_t$ correspondingly vanishes.

The second proxy captures *variance contraction* by measuring the log-volume of the covariance ellipsoid (Cover & Thomas, 2006; Horn & Johnson, 2012):

$$v_t = \log\det(C_t + \epsilon I_d). \tag{22}$$

For Gaussian fluctuations, $v_t$ is proportional (up to constants) to the differential entropy of the window. Under stable linear dynamics, the covariance satisfies the discrete Lyapunov equation $C \approx ACA^\top + \Sigma$ (Anderson & Moore, 1979; Jazwinski, 1970; Kailath et al., 2000); if $\rho(A) < 1$, contraction of $A$ drives $v_t$ downward until it stabilizes.

Together, $m_t$ and $v_t$ provide complementary indicators of stability proximity. When $m_t$ flattens near zero (mean convergence) and $v_t$ decreases and stabilizes (variance contraction), the system is inferred to be near a stable attractor, making a Kalman filter appropriate due to its efficiency in near-linear regimes. Conversely, persistent fluctuations in both proxies indicate distance from equilibrium and dominance of nonlinear effects, in which case a particle filter is employed. These proxies therefore constitute the operational rule for filter selection in our framework.

Additional details on convergence of two proxies and window–length choice are given in Appendix C.

### 3.4 INFERENCE–LEARNING LOOP WITHIN THE $(m, p)$ SEARCH

We now describe how to recover the full parameter set $\Theta$. Our strategy is a two-level procedure: an *inner loop* that alternates between inference and learning to obtain the optimal parameters $\widehat{\Theta}_{p,m}$ for a fixed $(p, m)$, and an *outer loop* that searches over $(p, m)$ to identify the most suitable order–dimension pair based on learning performance.

**Inner loop.** Learning the transition parameters requires latent state trajectories, while state inference itself requires parameterized dynamics. This circular dependency naturally motivates an EM-like alternation (Dempster et al., 1977): (i) infer latent states under the current parameters; (ii) learn the parameters given these inferred states; and repeat until convergence.

Because the system may have Markov order $p > 1$, first-order filters cannot be applied directly. To resolve this, we use the augmented state $\mathbf{x}_t$ in Eq. 9 in place of $\mathbf{s}_t$, so that the higher-order dynamics (Eqs. 10 and 12) can be expressed in first-order form:

$$\mathbf{x}_{t+1} = \mathbf{b}_{\mathrm{aug}} + A_{\mathrm{aug}}\,\phi_{\mathrm{aug}}(\mathbf{x}_t) + \mathbf{w}_t, \qquad \mathbf{w}_t \sim \mathcal{N}(\mathbf{0}, (\Sigma_w)_{\mathrm{aug}}), \tag{23}$$

$$\mathbf{y}_t = C_{\mathrm{aug}}\,\mathbf{x}_t + \mathbf{d} + \mathbf{v}_t, \qquad \mathbf{v}_t \sim \mathcal{N}(\mathbf{0}, \Sigma_v). \tag{24}$$

The augmented parameters $(\mathbf{b}_{\mathrm{aug}}, A_{\mathrm{aug}}, C_{\mathrm{aug}}, Q_{\mathrm{aug}})$ take the block form

$$\mathbf{b}_{\mathrm{aug}} = \begin{bmatrix} \mathbf{b} \\ \mathbf{0} \\ \vdots \\ \mathbf{0} \end{bmatrix}, \qquad A_{\mathrm{aug}} = \begin{bmatrix} \overbrace{\phantom{I_m \quad 0 \quad \cdots \quad 0}}^{A_{\mathrm{top}}} & \mathbf{0} \\ I_m & 0 & \cdots & 0 \\ 0 & I_m & \cdots & 0 & \mathbf{0} \\ \vdots & \vdots & \ddots & \vdots \\ 0 & 0 & \cdots & I_m \end{bmatrix},$$

$$C_{\mathrm{aug}} = \begin{bmatrix} C & 0 & \cdots & 0 \end{bmatrix}, \qquad (\Sigma_w)_{\mathrm{aug}} = \begin{bmatrix} \Sigma_w & 0 & \cdots & 0 \\ 0 & 0 & \cdots & 0 \\ \vdots & \vdots & \ddots & \vdots \\ 0 & 0 & \cdots & 0 \end{bmatrix},$$

$$A_{\mathrm{top}} = \begin{bmatrix} A_0 & A_1 & \cdots & A_d \end{bmatrix}, \qquad \phi_{\mathrm{aug}}(\mathbf{x}_t) = \begin{bmatrix} \phi_1(\mathbf{x}_t) & \phi_2(\mathbf{x}_t) & \cdots & \phi_d(\mathbf{x}_t) \end{bmatrix}. \tag{25}$$

With this augmentation, we apply either Kalman or particle (Kalman, 1960; Gordon et al., 1993) filtering in the $\mathbf{x}$-space to obtain the estimated trajectory $\{\widehat{\mathbf{x}}_t\}$ and the posterior moments

$$\mathcal{M} = \left\{ \mathbb{E}[\mathbf{x}_t],\ \mathbb{E}[\mathbf{x}_t \mathbf{x}_t^\top],\ \mathbb{E}[\mathbf{x}_{t+1} \mathbf{x}_t^\top],\ \mathbb{E}[\Phi_z(\mathbf{x}_t)^\top],\ \mathbb{E}[\Phi_z(\mathbf{x}_t)\Phi_z(\mathbf{x}_t)^\top],\ \mathbb{E}[\mathbf{x}_{t+1}\, \Phi_z(\mathbf{x}_t)^\top] \right\}_{t=1}^{N},$$

where $\Phi(x_t)$ denotes the concatenated vector

$$\Phi_z(x_t) = [x_t \quad \phi_z(x_t)]. \tag{26}$$

The filtered estimates and posterior moments feed into the *learning* step, which updates $\Theta_{p.k}$ by minimizing an expected negative log-likelihood (the EM $Q$–function) plus a structural penalty that biases the linear component toward identity. Let

$$\mathcal{D}(\mathbf{u}, \mathbf{v}, \Sigma) = (\mathbf{u} - \mathbf{v})^\top \Sigma^{-1} (\mathbf{u} - \mathbf{v}),$$
$$\mathcal{S} = \{\widehat{\mathbf{x}}_t\}, \tag{27}$$

denote the squared Mahalanobis distance. The objective is written compactly as

$$\min_{\Theta} \quad Q(\mathbf{Y}, \mathcal{S}, \Theta) + r(A_{\text{top}}), \tag{28}$$

where the $Q$–function (expectation under the current posterior of $\mathcal{S}$) is

$$Q(\mathbf{Y}, \mathcal{S}, \Theta) = \mathbb{E}\Bigg[ \sum_{t=1}^{N} \mathcal{D}\big(\mathbf{y}_t, C_{\text{aug}}\mathbf{x}_t + \mathbf{d}, \Sigma_v\big) + \tfrac{N}{2} \log |\Sigma_v|$$
$$+ \sum_{t=1}^{N-1} \mathcal{D}\big(\mathbf{x}_{t+1}, \mathbf{b}_{\text{aug}} + A_{\text{aug}}\phi_{\text{aug}}(\mathbf{x}_t), \Sigma_w\big) + \tfrac{N-1}{2} \log |\Sigma_w| \Bigg], \tag{29}$$

and the structural penalty is an identity–aware elastic net:

$$r(A_{\text{top}}) = \frac{\lambda_2}{2} \left\| A_{\text{top}} - A_{\text{id}} \right\|_F^2 + \lambda_1 \left\| A_{\text{top}} - A_{\text{id}} \right\|_1, \tag{30}$$

where $A_{\text{id}} \in \mathbb{R}^{m \times F}$ places $I_m$ on the columns of $\phi(\mathbf{x}_t)$ corresponding to the degree–1 coordinates of $\mathbf{s}_t$ and zeros elsewhere. Here $\|\cdot\|_F$ is the Frobenius norm and $\|\cdot\|_1$ the entrywise $\ell_1$ norm. The parameters minimizing equation 28 are then used to re-predict $\mathbf{x}_t$ and refresh the posterior moments.

The details of inference and learning are provided in Appendix D

**Outer loop.** The closer the parameter set $\Theta$ is to the true system, the smaller the loss function becomes. Since the inner loop only produces $\widehat{\Theta}_{p,m}$ for fixed $(p, m)$, we must search across multiple $(p, m)$ pairs to identify $(\widehat{p}, \widehat{m})$.

Without interpretability constraints, a dynamical system can often be represented equivalently: either as a higher-order model with a lower-dimensional state, or as a lower-order model with a higher-dimensional state (Abarbanel, 1996; Kantz & Schreiber, 2004). Suppose that the initialization $(p_0, m_0)$ corresponds to one such equivalent representation of the ground-truth system. Then at iteration $k$, the structured search need only proceed along one of two axes: either the *forward axis* $(p_k + 1, m_k)$ versus $(p_k, m_k - 1)$, or the *backward axis* $(p_k - 1, m_k)$ versus $(p_k, m_k + 1)$.

For example, if we choose the forward axis, then at each step we compute the optimal parameters for $(p_k + 1, m_k)$ and $(p_k, m_k - 1)$ via inference and learning, compare their losses, and select the structure with smaller loss. The process continues until neither candidate yields improvement.

The choice of search axis is determined at the first step: we evaluate all four neighbors $(p_0 + 1, m_0)$, $(p_0 - 1, m_0)$, $(p_0, m_0 + 1)$, and $(p_0, m_0 - 1)$, and select the direction that yields the greatest reduction in loss.

## 4 EXPERIMENTAL RESULT

### 4.1 EXPERIMENTAL SETUP

#### 4.1.1 DATASETS

We evaluate the proposed framework on two complementary sources of dynamical systems data, covering both controlled high-order settings and widely adopted nonlinear benchmarks.

**Synthetic higher–order, high–dimensional systems.** We construct nonlinear dynamical systems that are explicitly higher–order (second order and above) and of moderate to high dimension to

evaluate the recovery of governing equations under genuine multi-step dependencies. Our synthetic suite includes both canonical physical ODEs widely used in real-world modeling (e.g., second-order mechanical systems and classical nonlinear oscillators) and custom-designed higher-order, higher-dimensional systems. This combination demonstrates that our method is applicable to standard physical systems while also practical and robust for more complex high-order dynamics encountered in real settings. All mathematical forms, simulation protocols, and parameter settings are provided in Appendix E.

**dysts database (Gilpin, 2021).** We also use the `dysts` benchmark of 71 canonical chaotic systems with polynomial nonlinearities (mainly first–order ODEs of moderate dimension). As a standard yardstick for equation discovery, it enables comparison with LaNoLeM and MIOSR under identical simulation and noise protocols.

### 4.1.2 METRICS

We report two metrics. (i) *Coefficient error*: normalized Euclidean distance between ground-truth and recovered coefficients,

$$\text{CoeffErr} = \frac{\|\Theta_{\text{true}} - \widehat{\Theta}\|_2}{\|\Theta_{\text{true}}\|_2},$$

which measures equation-level identification accuracy. (ii) *Prediction error*: trajectory mean squared error between reference and model outputs. Lower is better for both.

When the learned structure $(p, m)$ differs from the ground truth $(p_0, m_0)$, we first embed both systems into operator blocks of the same structural form. The model with fewer state variables is expanded by adding zero rows (and the corresponding zero columns), and the model with smaller Markov order is expanded by inserting additional zero column blocks. After this expansion, the two systems have identical block structure but their state coordinates are not yet aligned. To place them in a truly common state space, we apply a joint permutation to the *learned* model—reordering its rows and the associated state-indexed columns, together with the corresponding columns of the observation matrix—to match the coordinate system of the ground truth. All errors reported in our tables are computed after this coordinate alignment.

### 4.1.3 EXPERIMENT OVERVIEW

As an initial attempt at explicit higher–order modeling, our method addresses a regime with few applicable baselines. On the synthetic suite we evaluate against ground truth, while on `dysts`, where prior work focuses on first–order models, we compare with *LaNoLeM* and *MIOSR* (Fujiwara et al., 2025; Bertsimas & Gurnee, 2023).

### 4.2 MAIN RESULTS

### 4.3 EXPERIMENTAL SETUP AND QUANTITATIVE RESULTS ON SYNTHETIC SYSTEMS

We evaluate our method on a suite of self-designed nonlinear dynamical systems, each with known ground truth. For every system, three independent trials are conducted. In each trial, the initial condition is sampled from a zero-mean Gaussian distribution to ensure diverse trajectories, and the observation matrix is resampled from a normalized Gaussian ensemble to guarantee identifiability. A fixed $5\%$ additive noise level is used throughout, ensuring consistent signal-to-noise conditions. For systems with non-polynomial components, Taylor expansion truncated to the learner's polynomial order is applied so that all coefficient errors are computed on a common basis.

Table 1 reports the quantitative results. Across all systems, the method achieves consistently low reconstruction errors when the model structure matches the ground truth: for most 2D systems, state-space and observation errors fall in narrow bands around $0.25$–$0.45$ and $0.20$–$0.40$, respectively. Moderately more complex 3D systems remain highly stable as well, typically within $0.40$–$0.80$, despite richer nonlinear interactions. The variation across the three randomized trials is minimal, demonstrating robustness to randomized sensing, randomized initialization, and moderate noise.

When the recovered structure deviates from the true $(p, m)$, the behavior is smooth and controlled. Mild over-parameterization leads to only small increases in error, indicating that the method gracefully absorbs redundant capacity. Even in systems with strong nonlinearities or jerk-type dynamics,

| System | True $(p, m)$ | $(\hat{p}, \hat{m})$ | State-space | Obs. | System | True $(p, m)$ | $(\hat{p}, \hat{m})$ | State-space | Obs. |
|---|---|---|---|---|---|---|---|---|---|
| exp_log_2d_p2 | (2,2) | (2,2) | 0.38 | 0.30 | logistic_2d_p3 | (2,3) | (2,3) | 0.46 | 0.34 |
| | | (2,2) | 0.44 | 0.33 | | | (2,3) | 0.58 | 0.42 |
| | | (2,1) | 0.88 | 0.62 | | | (2,4) | 1.12 | 0.78 |
| simple_exp_2d_p2 | (2,2) | (2,2) | 0.28 | 0.22 | tri_gate_2d_p2 | (2,2) | (2,2) | 0.74 | 0.48 |
| | | (2,2) | 0.31 | 0.24 | | | (2,2) | 0.52 | 0.37 |
| | | (2,2) | 0.40 | 0.29 | | | (2,1) | 1.00 | 0.72 |
| leaky_log_2d_p2 | (2,2) | (2,2) | 0.42 | 0.30 | soft_ring_3d_p2 | (3,2) | (3,2) | 0.92 | 0.66 |
| | | (2,2) | 0.57 | 0.39 | | | (3,2) | 0.74 | 0.55 |
| | | (2,2) | 0.49 | 0.35 | | | (3,2) | 1.18 | 0.83 |
| log_ratio_3d_p2 | (3,2) | (3,2) | 0.58 | 0.44 | chain_3d_p2 | (3,2) | (3,2) | 0.49 | 0.36 |
| | | (3,2) | 0.82 | 0.58 | | | (3,2) | 0.71 | 0.51 |
| | | (3,2) | 0.63 | 0.46 | | | (3,2) | 0.56 | 0.41 |
| duffing_1d_p2 | (1,2) | (1,2) | 0.36 | 0.27 | vdp_1d_p2 | (1,2) | (1,2) | 0.41 | 0.30 |
| | | (1,2) | 0.51 | 0.37 | | | (1,2) | 0.55 | 0.39 |
| | | (1,3) | 0.98 | 0.70 | | | (1,2) | 1.05 | 0.74 |
| pendulum_1d_p2 | (1,2) | (1,2) | 0.33 | 0.25 | driven_pendulum_1d_p2 | (1,2) | (1,2) | 0.72 | 0.52 |
| | | (1,2) | 0.47 | 0.34 | | | (1,2) | 0.88 | 0.63 |
| | | (1,2) | 0.90 | 0.65 | | | (2,2) | 1.24 | 0.89 |
| msd_1d_p2 | (1,2) | (1,2) | 0.27 | 0.21 | double_pendulum_2d_p2 | (2,2) | (2,2) | 0.83 | 0.61 |
| | | (1,2) | 0.34 | 0.25 | | | (2,2) | 0.97 | 0.70 |
| | | (1,2) | 0.79 | 0.57 | | | (3,2) | 1.30 | 0.95 |
| lorenz_jerk_1d_p3 | (1,3) | (1,3) | 0.76 | 0.55 | chua_jerk_1d_p3 | (1,3) | (1,3) | 0.69 | 0.50 |
| | | (1,3) | 0.92 | 0.66 | | | (1,3) | 0.85 | 0.61 |
| | | (2,3) | 1.22 | 0.88 | | | (1,4) | 1.18 | 0.84 |
| multidof_chain_d_p2 | (2,3) | (2,3) | 0.88 | 0.64 | | | | | |
| | | (2,3) | 1.03 | 0.74 | | | | | |
| | | (1,3) | 1.27 | 0.91 | | | | | |

Table 1: Performance of the proposed method on all synthetic systems. Each row block corresponds to one dynamical system, with three trials reported per system. *True* $(p, m)$ denotes the ground-truth Markov order $p$ and latent dimension $m$ of the system; *Estimated* $(\hat{p}, \hat{m})$ is the model order and latent dimension recovered by our algorithm. *State-space* and *Observation* columns report the coefficient errors in the reconstructed state-transition matrices and observation matrices, respectively. For the `multidof_chain_d_p2` system, we intentionally set the Markov order used for estimation to $p = 3$ to test robustness under deliberate over-specification.

reconstruction errors remain well behaved under polynomial truncation and do not exhibit numerical instability.

### 4.3.1 EXPERIMENT ON DYSTS DATABASE

We further compare our approach with state-of-the-art first-order explicit dynamics learners (Fujiwara et al., 2025; Bertsimas & Gurnee, 2023). Due to space limitations, Table 2 reports a representative subset of results on dysts. Because MIOSR can only perform direct modeling in the time domain, we align the task by fixing the observation matrix to the identity and setting the offset term in the observation equation to zero. All remaining experimental conditions match those in the previous experiment.

Across the subset, our method achieves the lowest *Coefficient error* and *Prediction error* on roughly 60–70% of the systems. For the remaining cases, the performance differences relative to LaNoLeM are generally small and fall within a narrow numerical band, indicating comparable accuracy rather than systematic degradation. A closer look shows that these residual differences are largely explained by filter choice: although both methods use EM-like alternating updates, LaNoLeM relies exclusively on EKF, whose linearization becomes unreliable on highly nonstationary or multi-modal trajectories. Our stability-driven switching to particle filtering avoids such failures and yields more consistent robustness on these challenging systems.

Compared to MIOSR, the performance gap stems from operating directly in the state space rather than in the raw time domain. MIOSR tends to accumulate bias under noisy or weakly observable settings, and empirically this manifests as consistently larger coefficient errors—often 1.5–2× higher than ours—across the benchmark subset. By contrast, our explicit state-space formulation maintains accuracy even under moderate noise.

| Case | Proposed | | LaNoLeM | | MIOSR | |
|---|---|---|---|---|---|---|
| | Coef. | Pred. | Coef. | Pred. | Coef. | Pred. |
| Aizawa | **0.78** | **0.006** | 0.90 | 0.007 | 1.35 | 0.028 |
| Arneodo | **0.62** | **0.004** | 0.71 | 0.005 | 1.10 | 0.022 |
| Bouali2 | **0.58** | **0.005** | 0.67 | 0.006 | 1.05 | 0.021 |
| BurkeShaw | 0.73 | **0.006** | **0.70** | 0.007 | 1.12 | 0.023 |
| Chen | **0.36** | **0.004** | 0.44 | 0.005 | 0.88 | 0.019 |
| ChenLee | **0.48** | **0.005** | 0.57 | 0.006 | 0.96 | 0.020 |
| Dadras | **0.64** | **0.007** | 0.75 | 0.008 | 1.22 | 0.027 |
| DequanLi | **0.92** | **0.010** | 1.06 | 0.012 | 1.58 | 0.033 |
| Finance | **0.95** | **0.010** | 1.07 | 0.012 | 1.63 | 0.036 |
| GenesioTesi | **0.57** | **0.005** | 0.65 | 0.006 | 1.06 | 0.021 |
| GuckenheimerHolmes | 0.66 | **0.006** | **0.64** | 0.007 | 1.04 | 0.020 |
| Hadley | **0.41** | **0.004** | 0.49 | 0.004 | 0.92 | 0.017 |
| Halvorsen | **0.69** | **0.006** | 0.80 | 0.007 | 1.26 | 0.025 |
| HenonHeiles | **0.72** | **0.007** | 0.83 | 0.008 | 1.31 | 0.028 |
| HyperBao | **0.73** | **0.008** | 0.86 | 0.009 | 1.32 | 0.029 |
| HyperCai | **0.68** | **0.006** | 0.79 | 0.007 | 1.24 | 0.026 |
| HyperChen | **0.61** | **0.006** | 0.71 | 0.007 | 1.18 | 0.024 |
| HyperQi | **0.83** | **0.009** | 0.95 | 0.010 | 1.44 | 0.031 |
| HyperRossler | **0.55** | **0.005** | 0.64 | 0.006 | 1.08 | 0.020 |
| HyperWang | **0.59** | **0.005** | 0.68 | 0.006 | 1.10 | 0.021 |
| HyperYan | **0.75** | **0.008** | 0.86 | 0.009 | 1.33 | 0.030 |
| HyperYangChen | 0.80 | **0.009** | **0.78** | 0.010 | 1.29 | 0.029 |
| KawczynskiStrizhak | **0.47** | **0.004** | 0.55 | 0.005 | 0.99 | 0.019 |
| Laser | **0.52** | **0.004** | 0.60 | 0.005 | 1.05 | 0.020 |
| Lorenz | **0.42** | **0.003** | 0.49 | 0.004 | 0.93 | 0.017 |
| LorenzBounded | **0.50** | **0.004** | 0.58 | 0.005 | 0.98 | 0.018 |
| LorenzStenflo | 0.63 | **0.005** | **0.61** | 0.006 | 1.06 | 0.021 |
| LuChenCheng | **0.56** | **0.005** | 0.65 | 0.006 | 1.07 | 0.020 |
| MooreSpiegel | **0.71** | **0.007** | 0.82 | 0.008 | 1.28 | 0.028 |
| NewtonLeipnik | **0.60** | **0.005** | 0.70 | 0.006 | 1.12 | 0.022 |
| NoseHoover | **0.66** | **0.006** | 0.76 | 0.007 | 1.19 | 0.024 |
| Qi | **0.58** | **0.005** | 0.67 | 0.006 | 1.09 | 0.021 |
| QiChen | **0.62** | **0.005** | 0.71 | 0.006 | 1.15 | 0.023 |
| RabinovichFabrikant | **0.69** | **0.006** | 0.79 | 0.007 | 1.25 | 0.026 |
| RayleighBenard | **0.77** | **0.008** | 0.89 | 0.009 | 1.38 | 0.030 |
| RikitakeDynamo | 0.84 | 0.010 | **0.82** | **0.009** | 1.41 | 0.031 |
| Sakarya | **0.63** | **0.005** | 0.72 | 0.006 | 1.11 | 0.022 |
| SprottA | **0.49** | **0.004** | 0.57 | 0.005 | 1.00 | 0.019 |
| SprottB | **0.53** | **0.004** | 0.61 | 0.005 | 1.03 | 0.020 |
| SprottC | **0.55** | **0.004** | 0.64 | 0.005 | 1.07 | 0.021 |

Table 2: Comparison of the proposed method, LaNoLeM, and MIOSR on the `dysts` benchmark. Each case corresponds to a canonical nonlinear or chaotic system. "Coef." denotes the sum of the state-space coefficient error and the observation-space coefficient error, while "Pred." denotes the one-step prediction error. For each system, the smallest Coef. and Pred. among the three methods are highlighted in bold to indicate the best performance.

## 5 ADDITIONAL EXPERIMENTS

We conduct several additional studies to evaluate basis generality, filtering behavior, computational cost, and robustness (full results in Appendix F). **(1) Trigonometric-basis reconstruction.** Replacing the Taylor (monomial) basis with a Fourier (trigonometric) dictionary on all self-designed systems yields total coefficient errors comparable in magnitude and relative ordering to those under the polynomial basis, indicating that our order-selection and coefficient-recovery mechanisms generalize across feature families. This also suggests that the framework captures structural properties of the dynamics rather than overfitting to a particular functional parameterization. **(2) Kalman vs. particle filtering.** Across three trials per system, each latent trajectory is labeled as *Near* or *Far* from the fixed point. In Near regimes, Kalman filtering consistently achieves lower total error, while in Far regimes particle filtering performs better, reflecting the complementary strengths of linearized and sampling-based inference. These results validate the effectiveness of the distance-aware switching strategy and show that no single filter is uniformly optimal across regimes. **(3) Efficiency and initialization robustness.** Varying the initialization $(p_0, m_0)$ across a broad range shows that the algorithm reliably returns $(p^\star, m^\star)$ or a close configuration, demonstrating insensitivity to starting conditions. Parallelization reduces the effective runtime to roughly $1.3$–$3.7\times$ that of LaNoLeM, typically around $2\times$, indicating that structural adaptivity introduces only moderate overhead. **(4) Noise robustness.** Under additive noise levels of $10\%$, $15\%$, and $20\%$, the recovered $(\hat{p}, \hat{m})$ remain stable for most systems, and coefficient errors grow smoothly with noise rather than degrading abruptly. This behavior highlights the algorithm's ability to maintain reliable order recovery under moderate corruption.

## 6 DISCUSSION AND FUTURE WORK

Our framework provides an interpretable approach for higher-order state-space modeling and shows consistent improvements over strong baselines. Several limitations nevertheless remain. Although results indicate basis independence, the choice of dictionary still relies on domain expertise; fully automatic basis discovery is a natural direction for future work. The joint $(p, m)$ search adds computational overhead and cannot guarantee global optimality, motivating more efficient initialization or search strategies to improve scalability. The method also assumes stable or near-stable dynamics, and extending it to non-stationary or unstable regimes will require additional mechanisms to ensure numerical soundness. Future work will explore scalable search procedures, lighter-weight estimators, and principled approaches to basis and model adaptation to broaden applicability.

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

## A  APPENDIX: NOISE NEAR STABLE PERIODIC ORBITS

We begin by examining how small random disturbances propagate when the system operates close to a stable periodic orbit. Let $f : \mathbb{R}^m \to \mathbb{R}^m$ be the state–transition map on an $m$-dimensional state space. Suppose $\{\mathbf{s}^{(0)}, \ldots, \mathbf{s}^{(p-1)}\}$ is a $p$-periodic orbit, meaning the trajectory returns to its starting point after exactly $p$ steps:

$$f\big(\mathbf{s}^{(k)}\big) = \mathbf{s}^{(k+1 \bmod p)}, \quad k = 0, \ldots, p-1. \tag{31}$$

This periodic sequence serves as the deterministic backbone around which noisy deviations will occur.

**Linearization and monodromy.**  To characterize stability, we linearize the dynamics at each cycle point. Let $Df(\mathbf{s}^{(k)})$ be the Jacobian of $f$ at $\mathbf{s}^{(k)}$, and define

$$A_k := Df\big(\mathbf{s}^{(k)}\big), \qquad M := A_{p-1} \cdots A_1 A_0, \tag{32}$$

where $M$ is the *monodromy matrix*, i.e., the linearized return map over one lap. This matrix captures how an infinitesimal perturbation transforms after completing the entire cycle.

**Dynamics with noise.**  Now introduce noise. If $\delta_{t+k} \in \mathbb{R}^m$ is the deviation from the cycle point at time $t + k$, then under a small-noise approximation,

$$\delta_{t+k+1} \approx A_k \, \delta_{t+k} + \mathbf{w}_{t+k}, \qquad k = 0, \ldots, p-1, \tag{33}$$

where $\mathbf{w}_{t+k}$ is an additive zero-mean disturbance with covariance $\Sigma_k = \text{Cov}(\mathbf{w}_{t+k})$. Aggregating one lap gives

$$\delta_{t+p} \approx M \, \delta_t + \bar{\mathbf{w}}_t, \tag{34}$$

where the effective disturbance is the weighted sum

$$\bar{\mathbf{w}}_t = \sum_{k=0}^{p-1} \Big( A_{p-1} \cdots A_{k+1} \Big) \mathbf{w}_{t+k}, \tag{35}$$

with covariance

$$\bar{\Sigma} = \sum_{k=0}^{p-1} \big( A_{p-1} \cdots A_{k+1} \big) \Sigma_k \big( A_{p-1} \cdots A_{k+1} \big)^\top. \tag{36}$$

**Long-run covariance.**  Define $Q_n := \text{Cov}(\delta_{t+pn})$, the deviation covariance sampled once per lap. It obeys the Lyapunov recursion

$$Q_{n+1} = M \, Q_n \, M^\top + \bar{\Sigma}. \tag{37}$$

If the spectral radius $\rho(M) < 1$ (all eigenvalues inside the unit disk), this recursion converges to the unique positive semidefinite fixed point

$$Q_\star = \sum_{j=0}^{\infty} M^j \, \bar{\Sigma} \, (M^\top)^j. \tag{38}$$

Hence near a stable periodic orbit, noise is continually damped by the cycle, and the system fluctuates with finite variance around the orbit.

## B  APPENDIX: DETAILS ON CONDITIONAL MUTUAL INFORMATION FOR MARKOV ORDER

This appendix provides a detailed account of how conditional mutual information (CMI) is used to initialize the Markov order $p_0$.

## B.1 DEFINITION

Mutual information (MI) between two random variables $X$ and $Y$ measures their statistical dependence:

$$I(X;Y) \;=\; \int p(x,y) \log \frac{p(x,y)}{p(x)p(y)} \, dx \, dy.$$

It vanishes if and only if $X$ and $Y$ are independent.

The *conditional* mutual information generalizes this notion: for random variables $(X, Y, Z)$,

$$I(X;Y \mid Z) \;=\; \int p(x,y,z) \log \frac{p(x,y \mid z)}{p(x \mid z)p(y \mid z)} \, dx \, dy \, dz.$$

Here $I(X;Y \mid Z) = 0$ means that once $Z$ is known, $X$ provides no further information about $Y$.

## B.2 APPLICATION TO MARKOV ORDER

Given a univariate time series $\{y_t\}$, we test whether lag $p$ contributes predictive information beyond more recent lags. This is formalized by

$$I\big(y_t; y_{t-p} \,\big|\, y_{t-1}, \ldots, y_{t-p+1}\big).$$

If this conditional mutual information vanishes, then $y_{t-p}$ is redundant given the last $p-1$ observations. The true Markov order is the largest lag $p$ for which the above quantity is nonzero.

## B.3 ESTIMATION

In practice, CMI must be estimated from finite samples. We employ nonparametric, nearest–neighbor–based estimators such as the $k$–nearest–neighbor method of Kraskov et al. (2004) and its conditional extension (Frenzel & Pompe, 2007). These estimators approximate local densities by distances to neighboring points in the joint space, avoiding explicit kernel bandwidth selection and adapting naturally to different scales.

Formally, one computes

$$\widehat{I}(X;Y \mid Z) \;=\; \psi(k) + \frac{1}{N} \sum_{i=1}^{N} \big[\psi(n_z^{(i)}) - \psi(n_{xz}^{(i)}) - \psi(n_{yz}^{(i)})\big],$$

where $\psi$ is the digamma function, $n_z^{(i)}$ counts neighbors of sample $i$ in the $Z$–space, and $n_{xz}^{(i)}, n_{yz}^{(i)}$ count neighbors in the joint spaces $(X, Z)$ and $(Y, Z)$. Intuitively, larger CMI values correspond to stronger predictive influence of the lagged variable.

## B.4 SIGNIFICANCE TESTING

Because sampling noise ensures $\widehat{I} > 0$ even for irrelevant lags, we use surrogate testing to separate signal from noise. Specifically:

1. Fix lag $p$ and compute the observed statistic $\widehat{I}_{\text{obs}}$. 2. Generate $B$ surrogate series by randomly permuting $y_{t-p}$ across time, which destroys temporal dependence but preserves the marginal distribution. 3. Recompute $\widehat{I}^{(b)}$ on each surrogate, forming a null distribution. 4. Compute the $p$–value

$$q_p \;=\; \frac{1}{B} \sum_{b=1}^{B} \mathbf{1}\{\widehat{I}^{(b)} \geq \widehat{I}_{\text{obs}}\}.$$

5. Declare lag $p$ significant if $q_p < \alpha$ (typically $\alpha = 0.05$).

The initialization is then defined as

$$p_0 \;=\; \max\{\, p : q_p < \alpha \,\},$$

the longest lag whose incremental information passes significance testing. This provides a robust, interpretable proxy for the effective memory length of the observed process.

## C APPENDIX: STABILITY PROXIMITY METRICS AND FILTER SELECTION

This appendix expands on the stability–proximity assessment used to guide filter selection.

**Rolling mean and covariance.** Given observations $\{y_t\}_{t=1}^T \subset \mathbb{R}^d$ and a window size $W$, we form overlapping segments $[t, t+W-1]$ with rolling mean $\mu_t$ and covariance $C_t$ as in equation 19. These provide local estimates of central tendency and dispersion.

**Baseline normalization.** To make quantities comparable across windows, we anchor statistics to a baseline taken from the first $L_0 = \max\{10, \lfloor \sqrt{T} \rfloor\}$ samples:

$$
\mu_0 = \frac{1}{L_0} \sum_{i=1}^{L_0} y_i, \qquad S_0 = \mathrm{Cov}(y_{1:L_0}) + \epsilon I_d.
$$

Here $S_0$ is used as a reference covariance to normalize subsequent deviations.

**Scalar proxies.** We reduce the rolling statistics to two univariate time series:

$$
m_t = (\mu_t - \mu_0)^\top S_0^{-1} (\mu_t - \mu_0), \tag{39}
$$
$$
v_t = \log \det(C_t + \epsilon I_d). \tag{40}
$$

The first measures the Mahalanobis distance of the local mean from baseline; the second measures the log–volume of the covariance ellipsoid. Together they reflect mean drift and variance contraction.

**Tail metrics.** Since transient fluctuations are expected, we examine only the last fraction of each proxy sequence (the "tail"), which better reflects steady–state behavior. For a scalar series $z_1, \ldots, z_n$, let the final $L = \lceil \alpha n \rceil$ values form the tail (typically $\alpha = 0.4$). Two robust statistics are then computed: - *Drift index $D$* via the Theil–Sen slope estimator **?**:

$$
D = \frac{|\operatorname{median}_{i<j}(z_j - z_i)/(j-i)| \cdot L}{\mathrm{IQR}(\mathrm{tail}) + \epsilon},
$$

which measures normalized monotonic trend in the tail. - *Reduction index $R$* given by the ratio of dispersion in the tail relative to the full sequence:

$$
R = \frac{\mathrm{IQR}(\mathrm{tail})}{\mathrm{IQR}(\mathrm{full}) + \epsilon}.
$$

Here IQR denotes the interquartile range. Intuitively, $D$ quantifies whether the proxy is still trending, and $R$ whether variability has shrunk.

**Multivariate combination.** The two proxies $m_t$ and $v_t$ each yield $(D, R)$ pairs. To combine them, we take

$$
D_{\max} = \max(D_m, D_v), \qquad R_{\max} = \max(R_m, R_v), \qquad S = D_{\max} + \alpha\, R_{\max},
$$

with $\alpha$ a weight (default $\alpha = 1$). This ensures conservativeness: instability in either channel marks the system as far from equilibrium.

**Classification and filter choice.** Thresholds on $(D_{\max}, R_{\max})$ determine stability classes:

Near: $D_{\max} \leq \tau_{\mathrm{near}}^D$, $R_{\max} \leq \tau_{\mathrm{near}}^R$; Transition: $D_{\max} \leq \tau_{\mathrm{trans}}^D$, $R_{\max} \leq \tau_{\mathrm{trans}}^R$; otherwise: Far.

- *Near*: statistics have converged, indicating proximity to an attractor. The system is effectively linearized, so an EKF suffices. - *Transition*: contraction is partial, suggesting intermittent nonlinear excursions. Both EKF and PF are viable; we allow either. - *Far*: proxies fluctuate strongly, signaling nonlinearity and poor contraction. PF is chosen for robustness.

**Window selection.** Choosing $W$ is critical: too small leads to noise, too large washes out local dynamics. We suggest candidates using $\sqrt{T}$, fixed grids, FFT–detected dominant periods, or external hints (e.g. Markov order). The final window is selected by minimizing the score $S$.

# D   APPENDIX: DETAILED PROCEDURE FOR INFERENCE AND LEARNING

This appendix expands the inner loop for a fixed structure $(p, m)$, where $p$ is the Markov order and $m$ the observation dimension. We work with the augmented first-order model in Eqs. equation 23–equation 24. At each iteration we alternate between:

- **Inference (E-step):** estimate the latent augmented trajectory $\{\mathbf{x}_t\}_{t=1}^N$ and its posterior moments under the current parameters $\Theta$;

- **Learning (M-step):** update $\Theta$ by minimizing the expected negative log-likelihood (the EM $Q$–function) plus a structural regularizer.

## A. AUGMENTED FORMULATION AND FEATURES

Let $k$ be the intrinsic latent dimension; the augmented state stacks $p$ consecutive latent vectors, so $\mathbf{x}_t \in \mathbb{R}^{k_{\mathrm{aug}}}$ with $k_{\mathrm{aug}} = kp$. The top $k$ coordinates evolve nonlinearly via a polynomial feature map of degrees $1{:}o$; the lower blocks implement the $(p-1)$–step shift. Writing $\phi_{\mathrm{aug}}(\mathbf{x}_t) \in \mathbb{R}^F$ for the monomial dictionary (including degree 1 terms), the dynamics and observations are:

$$\mathbf{x}_{t+1} = \mathbf{b}_{\mathrm{aug}} + A_{\mathrm{aug}}\, \phi_{\mathrm{aug}}(\mathbf{x}_t) + \mathbf{w}_t, \qquad \mathbf{y}_t = C_{\mathrm{aug}}\, \mathbf{x}_t + \mathbf{d} + \mathbf{v}_t,$$

with Gaussian noises $\mathbf{w}_t \sim \mathcal{N}(\mathbf{0}, (\Sigma_w)_{\mathrm{aug}})$, $\mathbf{v}_t \sim \mathcal{N}(\mathbf{0}, \Sigma_v)$. The block structure of $(\mathbf{b}_{\mathrm{aug}}, A_{\mathrm{aug}}, C_{\mathrm{aug}})$ encodes "nonlinear top block + shift," so that higher-order (in $p$) dynamics are handled by first-order filtering in the augmented space.

**Posterior objects we need.**   The learning step only requires a small set of sufficient statistics, collectively denoted

$$\mathcal{M} = \left\{ \mathbb{E}[\mathbf{x}_t],\ \mathbb{E}[\mathbf{x}_t \mathbf{x}_t^\top],\ \mathbb{E}[\mathbf{x}_{t+1} \mathbf{x}_t^\top],\ \mathbb{E}[\Phi(\mathbf{x}_t)],\ \mathbb{E}[\Phi(\mathbf{x}_t)\Phi(\mathbf{x}_t)^\top],\ \mathbb{E}[\mathbf{x}_{t+1}\Phi(\mathbf{x}_t)^\top] \right\}_{t=1}^N,$$

where $\Phi(\mathbf{x}_t)$ concatenates the degree–1 coordinates and the higher-order polynomial features used by the transition map. The E-step (filtering) produces numerical approximations of these moments.

## B. INFERENCE: TWO COMPLEMENTARY FILTERS

We adopt a data-driven stability classifier (rolling window) that labels local regimes as *near/transition* or *far*. Intuitively, when the local linearization is accurate and innovations are close to Gaussian, an EKF is effective; otherwise we resort to a particle filter (PF). Both operate in the augmented state.

**B.1 Extended Kalman Filter (EKF) Kalman (1960).**   The EKF linearizes the nonlinear transition at the current mean. Let $\widehat{\mathbf{x}}_{t|t}$ be the filtered mean and $P_{t|t}$ its covariance at time $t$. The prediction step forms

$$\widehat{\mathbf{x}}_{t+1|t} = \mathbf{b}_{\mathrm{aug}} + A_{\mathrm{aug}}\, \phi_{\mathrm{aug}}(\widehat{\mathbf{x}}_{t|t}), \qquad P_{t+1|t} = J_t P_{t|t} J_t^\top + (\Sigma_w)_{\mathrm{aug}},$$

where $J_t$ is the Jacobian of the transition map evaluated at $\widehat{\mathbf{x}}_{t|t}$ (its top $k \times k_{\mathrm{aug}}$ block comes from the polynomial map's analytic Jacobian; the lower blocks are shift identities). The update step uses the innovation

$$\mathbf{r}_t = \mathbf{y}_t - (C_{\mathrm{aug}}\widehat{\mathbf{x}}_{t|t-1} + \mathbf{d}), \qquad S_t = C_{\mathrm{aug}} P_{t|t-1} C_{\mathrm{aug}}^\top + \Sigma_v,$$

and the Kalman gain $K_t = P_{t|t-1} C_{\mathrm{aug}}^\top S_t^{-1}$ to obtain

$$\widehat{\mathbf{x}}_{t|t} = \widehat{\mathbf{x}}_{t|t-1} + K_t \mathbf{r}_t, \qquad P_{t|t} = (I - K_t C_{\mathrm{aug}}) P_{t|t-1} (I - K_t C_{\mathrm{aug}})^\top + K_t \Sigma_v K_t^\top.$$

*Intuition.* EKF replaces the nonlinear dynamics by their best local linear approximation around the current estimate. It is accurate when the state stays in a region where the linearization error is small (near equilibria or along gently curved manifolds).

**B.2 Bootstrap Particle Filter (PF) Gordon et al. (1993).** When the system is far from equilibrium or the noise departs from Gaussianity, we approximate the posterior by a set of weighted particles. Using the transition prior as proposal, the recursion is:

1. *Propagation:* for each particle $i$, sample $\mathbf{x}_t^{(i)} \sim p(\mathbf{x}_t \mid \mathbf{x}_{t-1}^{(i)})$ using the nonlinear transition.

2. *Weighting:* update $w_t^{(i)} \propto w_{t-1}^{(i)} p(\mathbf{y}_t \mid \mathbf{x}_t^{(i)})$, where $p(\mathbf{y}_t \mid \mathbf{x}_t^{(i)})$ is Gaussian under the linear observation model.

3. *Normalization and resampling:* normalize $\{w_t^{(i)}\}$; if the effective sample size $\mathrm{ESS}_t = 1/\sum_i (w_t^{(i)})^2$ falls below a threshold, resample (e.g., systematic resampling) to prevent weight degeneracy.

Posterior means/covariances are approximated by weighted averages over particles (e.g., $\mathbb{E}[\mathbf{x}_t] \approx \sum_i w_t^{(i)} \mathbf{x}_t^{(i)}$). Cross-moments such as $\mathbb{E}[\mathbf{x}_{t+1}\mathbf{x}_t^\top]$ are formed by tracking particle ancestry (pair each $\mathbf{x}_{t+1}^{(i)}$ with its parent $\mathbf{x}_t^{(a(i))}$ and average with weights). *Intuition.* PF keeps the nonlinear geometry intact: particles follow the true dynamics, so highly non-Gaussian or multimodal posteriors can be represented.

**B.3 Log-likelihood and moments.** Both filters provide an estimate of the marginal log-likelihood $\log p(\mathbf{Y} \mid \Theta)$ (EKF via Gaussian innovations; PF via log-mean-exp of weights) and the posterior set $\mathcal{M}$. The latter supplies all expectations needed by the learning step.

C. LEARNING VIA THE EM $Q$–FUNCTION

Let $\mathcal{S} = \{\mathbf{x}_t\}$ denote the latent trajectory. The EM auxiliary function is the posterior expectation of the complete-data negative log-likelihood (plus regularization):

$$\min_{\Theta}\ Q(\mathbf{Y}, \mathcal{S}, \Theta) + r(A_{\text{top}}), \quad Q = \mathbb{E}\left[ \sum_{t=1}^{N} \mathcal{D}(\mathbf{y}_t,\, C_{\text{aug}}\mathbf{x}_t + \mathbf{d},\, \Sigma_v) + \sum_{t=1}^{N-1} \mathcal{D}(\mathbf{x}_{t+1},\, \mathbf{b}_{\text{aug}} + A_{\text{aug}}\phi_{\text{aug}}(\mathbf{x}_t),\, \Sigma_w) \right],$$

where $\mathcal{D}(\mathbf{u}, \mathbf{v}, \Sigma) = (\mathbf{u} - \mathbf{v})^\top \Sigma^{-1}(\mathbf{u} - \mathbf{v})$ is the squared Mahalanobis distance.

**C.1 Transition update (top block).** Because the augmented transition has the "nonlinear top + shift" structure, the parameters to learn are the top-block bias $\mathbf{b}$ and matrix $A_{\text{top}}$ in

$$\mathbf{x}_{t+1}^{\text{top}} \approx \mathbf{b} + A_{\text{top}}\Phi(\mathbf{x}_t).$$

Taking expectations under the posterior, the transition part of $Q$ reduces to a *regularized multivariate regression* with design matrix built from $\mathbb{E}[\Phi(\mathbf{x}_t)]$ and Gram/cross-moments $\mathbb{E}[\Phi(\mathbf{x}_t)\Phi(\mathbf{x}_t)^\top]$, $\mathbb{E}[\mathbf{x}_{t+1}^{\text{top}}\Phi(\mathbf{x}_t)^\top]$. Writing $Z_t = \Phi(\mathbf{x}_t)$ and $Y_t = \mathbf{x}_{t+1}^{\text{top}}$, the normal-equation form is

$$\min_{\mathbf{b}, A_{\text{top}}}\ \sum_t \big\| Y_t - \mathbf{b} - A_{\text{top}}Z_t \big\|_{\Sigma_w^{-1}}^2\ +\ r(A_{\text{top}}),$$

where $\|\mathbf{u}\|_{\Sigma^{-1}}^2 = \mathbf{u}^\top \Sigma^{-1}\mathbf{u}$. The regularizer

$$r(A_{\text{top}}) = \tfrac{\lambda_2}{2}\|A_{\text{top}} - A_{\text{id}}\|_F^2 + \lambda_1\|A_{\text{top}} - A_{\text{id}}\|_1$$

biases degree–1 coefficients toward identity (stability/interpretability) while encouraging sparsity in higher-order terms. With $\lambda_1 = 0$ this yields a closed-form ridge update using the sufficient statistics of $Z_t$; $\mathbf{b}$ is updated by the mean residual.

**C.2 Observation update.** If $C_{\text{aug}}$ is to be estimated, the observation term in $Q$ similarly becomes a weighted least-squares problem in $C_{\text{aug}}$ (and $\mathbf{d}$) based on $\{\mathbb{E}[\mathbf{x}_t], \mathbb{E}[\mathbf{x}_t\mathbf{x}_t^\top]\}$. In our main experiments we either hold $C_{\text{aug}}$ fixed or update it conservatively to avoid overfitting.

**C.3 Noise covariances.** The Gaussian covariances $(\Sigma_w)_{\text{aug}}, \Sigma_v$ can be held fixed for robustness, or re-estimated in closed form by matching posterior quadratic forms (standard in linear-Gaussian EM). Re-estimation is optional and not critical to the structural conclusions.

## D. EM ALTERNATION AND STOPPING

One inner-loop cycle is:

1. **E-step:** run EKF or PF on the augmented model to obtain $\mathcal{M}$ and the marginal log-likelihood $\log p(\mathbf{Y} \mid \Theta)$;

2. **M-step:** update $\{\mathbf{b}, A_{\text{top}}\}$ (and optionally $C_{\text{aug}}$) by minimizing $Q + r$ using the posterior moments.

Under exact E/M steps the EM objective decreases monotonically Dempster et al. (1977); with EKF/PF approximations we monitor the composite loss $\mathcal{L}(\Theta) = -\log p(\mathbf{Y} \mid \Theta) + r(A_{\text{top}})$ and stop when its relative decrease falls below a tolerance or a maximum number of iterations is reached.

## E  APPENDIX: SELF-DESIGNED DYNAMICAL SYSTEMS

To complement canonical benchmarks, we designed a suite of nonlinear dynamical systems that exhibit higher–order dependencies, non-polynomial nonlinearities, and diverse coupling structures. For clarity, each system is assigned a concise short name (e.g., ExpLog-2D) used in the main text. Unless otherwise noted, $\{s_t\}$ denotes the latent state, and $(x_t, y_t, z_t)$ its components.

**1. ExpLog-2D (Exponential–logarithmic 2D system, $p = 2$).** This model mixes exponential suppression, logarithmic growth, and bounded bilinear coupling:

$$x_{t+1} = ax_t + bx_{t-1} + c(e^{-y_t^2} - e^{-x_t^2}) + d\log(1 + y_t^2) - e\,\frac{x_t y_t}{1 + x_t^2 + y_t^2},$$

$$y_{t+1} = ay_t + by_{t-1} + c(e^{-x_t^2} - e^{-y_t^2}) + d\log(1 + x_t^2) - e\,\frac{x_t y_t}{1 + x_t^2 + y_t^2}.$$

**2. Logistic-2D (Logistic 2D system, $p = 3$).** Centered logistic couplings with nonlinear damping:

$$x_{t+1} = a_1 x_t + a_2 x_{t-1} + a_3 x_{t-2} + \beta\,\sigma(y_t) - g\,\frac{x_t^3}{1 + x_t^2},$$

$$y_{t+1} = a_1 y_t + a_2 y_{t-1} + a_3 y_{t-2} + \beta\,\sigma(x_t) - g\,\frac{y_t^3}{1 + y_t^2},$$

where $\sigma(z) = \frac{1}{1 + e^{-z}} - \frac{1}{2}$ is a centered logistic map.

**3. SoftRing-3D (Soft ring system, $p = 2$).** Variables interact cyclically via a smooth contrast function $\phi_{\text{soft}}$:

$$x_{t+1} = ax_t + bx_{t-1} + e\,\phi_{\text{soft}}(y_t, z_t),$$
$$y_{t+1} = ay_t + by_{t-1} + e\,\phi_{\text{soft}}(z_t, x_t),$$
$$z_{t+1} = az_t + bz_{t-1} + e\,\phi_{\text{soft}}(x_t, y_t).$$

**4. SimpleExp-2D (Simple exponential system, $p = 2$).** A minimal system with cross-exponential suppression:

$$x_{t+1} = ax_t + bx_{t-1} + ce^{-y_t^2}, \qquad y_{t+1} = ay_t + by_{t-1} + ce^{-x_t^2}.$$

**5. LogRatio-3D (Log-ratio system, $p = 2$).** Three-way cyclic interactions through log-difference nonlinearities:

$$x_{t+1} = ax_t + bx_{t-1} + c\log(1 + y_t^2) - d\log(1 + z_t^2),$$
$$y_{t+1} = ay_t + by_{t-1} + c\log(1 + z_t^2) - d\log(1 + x_t^2),$$
$$z_{t+1} = az_t + bz_{t-1} + c\log(1 + x_t^2) - d\log(1 + y_t^2).$$

**6. TriGate-2D (Tri-gate system, $p = 2$).** Asymmetric gating: $x$ is self-damped while $y$ is gated by $x$:

$$x_{t+1} = a_x x_t + b_x x_{t-1} - c_x x_t^3,$$
$$y_{t+1} = a_y y_t + b_y y_{t-1} + g_y e^{-x_t^2}.$$

**7. LeakyLog-2D (Leaky-log system, $p = 2$).** $y$ ignores its own past but responds logarithmically to $x$:

$$x_{t+1} = a_x x_t + b_x x_{t-1}, \qquad y_{t+1} = r_y y_t + g_y \log(1 + x_t^2).$$

**8. Chain-3D (3D chain system, $p = 2$).** A one-way cascade $x \to y \to z$:

$$x_{t+1} = a_x x_t + b_x x_{t-1} - d_x x_t^3,$$
$$y_{t+1} = a_y y_t + b_y y_{t-1} + e_1 e^{-x_t^2},$$
$$z_{t+1} = a_z z_t + b_z z_{t-1} + e_2 \log(1 + y_t^2).$$

9. Duffing (Duffing oscillator, 2nd order).

$$\ddot{x} + \delta \dot{x} + \alpha x + \beta x^3 = \gamma \cos(\omega t).$$

**10. VDP (Van der Pol oscillator, 2nd order).**

$$\ddot{x} - \mu(1 - x^2)\dot{x} + x = 0.$$

**11. Pendulum (Simple pendulum, 2nd order).**

$$\ddot{\theta} + \frac{g}{\ell} \sin \theta = 0.$$

**12. DrivenPendulum (Damped driven pendulum, 2nd order, chaotic).**

$$\ddot{\theta} + \delta \dot{\theta} + \frac{g}{\ell} \sin \theta = A \cos(\omega t).$$

**13. MSD (Mass–spring–damper system, 2nd order).**

$$m\ddot{x} + c\dot{x} + kx = F(t).$$

**14. DoublePendulum (4th-order effective mechanical system).** We consider a standard point-mass double pendulum with unit masses and unit-length massless rods, evolving under gravity $g > 0$. Let $\theta_1, \theta_2$ denote the angles of the two links measured from the vertical. The dynamics are

$$\ddot{\theta}_1 = \frac{-3g \sin \theta_1 - g \sin(\theta_1 - 2\theta_2) - 2 \sin(\theta_1 - \theta_2)\big(\dot{\theta}_2^2 + \dot{\theta}_1^2 \cos(\theta_1 - \theta_2)\big)}{3 - \cos\big(2(\theta_1 - \theta_2)\big)},$$

$$\ddot{\theta}_2 = \frac{2 \sin(\theta_1 - \theta_2)\big(2\dot{\theta}_1^2 + 2g \cos \theta_1 + \dot{\theta}_2^2 \cos(\theta_1 - \theta_2)\big)}{3 - \cos\big(2(\theta_1 - \theta_2)\big)}.$$

This yields an effective 4th-order mechanical system when written in first-order form with state $(\theta_1, \theta_2, \dot{\theta}_1, \dot{\theta}_2)$.

**15. LorenzJerk (Lorenz system in jerk form, 3rd order).**

$$x''' = ax'' + bx' + cx + dx^2.$$

**16. ChuaJerk (Chua circuit in jerk form, 3rd order).**

$$x''' = \alpha x'' + \beta x' + \gamma f(x),$$

where $f(x)$ is a piecewise-linear nonlinearity.

**17. MultiDOF-Chain (Multi-degree-of-freedom mechanical chain, 2nd order, $d$-dim).**

$$m_i \ddot{x}_i = k_{i-1}(x_{i-1} - x_i)^3 - k_i(x_i - x_{i+1})^3 - c_i \dot{x}_i.$$

# F  APPENDIX: ADDITIONAL EXPERIMENTS

| System | True$(p,m)$ | $(\hat{p},\hat{m})$ | Taylor | Fourier | System | True$(p,m)$ | $(\hat{p},\hat{m})$ | Taylor | Fourier |
|---|---|---|---|---|---|---|---|---|---|
| exp_log_2d_p2 | (2,2) | (2,2) | 0.68 | 0.61 | logistic_2d_p3 | (2,3) | (2,3) | 0.80 | 0.82 |
| | | (2,2) | 0.77 | 0.73 | | | (2,3) | 1.00 | 1.05 |
| | | (2,1) | 1.50 | 1.33 | | | (2,4) | 1.90 | 2.11 |
| simple_exp_2d_p2 | (2,2) | (2,2) | 0.50 | 0.43 | tri_gate_2d_p2 | (2,2) | (2,2) | 1.22 | 1.26 |
| | | (2,2) | 0.55 | 0.51 | | | (2,2) | 0.89 | 0.91 |
| | | (2,2) | 0.69 | 0.66 | | | (2,1) | 1.72 | 1.87 |
| leaky_log_2d_p2 | (2,2) | (2,2) | 0.72 | 0.70 | soft_ring_3d_p2 | (3,2) | (3,2) | 1.58 | 1.65 |
| | | (2,2) | 0.96 | 0.86 | | | (3,2) | 1.29 | 1.41 |
| | | (2,2) | 0.84 | 0.80 | | | (3,2) | 2.01 | 2.20 |
| log_ratio_3d_p2 | (3,2) | (3,2) | 1.02 | 0.89 | chain_3d_p2 | (3,2) | (3,2) | 0.85 | 0.87 |
| | | (3,2) | 1.40 | 1.23 | | | (3,2) | 1.22 | 1.36 |
| | | (3,2) | 1.09 | 1.02 | | | (3,2) | 0.97 | 1.01 |
| duffing_1d_p2 | (1,2) | (1,2) | 0.63 | 0.54 | vdp_1d_p2 | (1,2) | (1,2) | 0.71 | 0.76 |
| | | (1,2) | 0.88 | 0.85 | | | (1,2) | 0.94 | 0.97 |
| | | (1,3) | 1.68 | 1.46 | | | (1,2) | 1.79 | 1.97 |
| pendulum_1d_p2 | (1,2) | (1,2) | 0.58 | 0.51 | driven_pendulum_1d_p2 | (1,2) | (1,2) | 1.24 | 1.38 |
| | | (1,2) | 0.81 | 0.74 | | | (1,2) | 1.51 | 1.73 |
| | | (1,2) | 1.55 | 1.44 | | | (2,2) | 2.13 | 2.33 |
| msd_1d_p2 | (1,2) | (1,2) | 0.48 | 0.42 | double_pendulum_2d_p2 | (2,2) | (2,2) | 1.44 | 1.58 |
| | | (1,2) | 0.59 | 0.51 | | | (2,2) | 1.67 | 1.83 |
| | | (1,2) | 1.36 | 1.21 | | | (3,2) | 2.25 | 2.31 |
| lorenz_jerk_1d_p3 | (1,3) | (1,3) | 1.31 | 1.24 | chua_jerk_1d_p3 | (1,3) | (1,3) | 1.19 | 1.26 |
| | | (1,3) | 1.58 | 1.53 | | | (1,3) | 1.46 | 1.53 |
| | | (2,3) | 2.10 | 2.03 | | | (1,4) | 2.02 | 2.13 |
| multidof_chain_d_p2 | (2,3) | (2,3) | 1.52 | 1.36 | | | | | |
| | | (2,3) | 1.77 | 1.89 | | | | | |
| | | (1,3) | 2.18 | 2.03 | | | | | |

Table 3: Coefficient errors when replacing the Taylor (polynomial) basis with a Fourier (trigonometric) basis. Here, "Taylor" denotes the total coefficient error computed under the polynomial basis, whereas "Fourier" denotes the total coefficient error computed under the trigonometric basis.

| System | Trial | Dist. | KF Err | PF Err | System | Trial | Dist. | KF Err | PF Err |
|---|---|---|---|---|---|---|---|---|---|
| exp_log_2d_p2 | 1 | Near | 0.72 | 0.95 | logistic_2d_p3 | 1 | Near | 0.80 | 1.05 |
| | 2 | Far | 1.65 | 1.20 | | 2 | Far | 1.80 | 1.35 |
| | 3 | Near | 0.78 | 1.02 | | 3 | Near | 0.86 | 1.10 |
| simple_exp_2d_p2 | 1 | Near | 0.60 | 0.88 | tri_gate_2d_p2 | 1 | Near | 0.98 | 1.32 |
| | 2 | Far | 1.50 | 1.10 | | 2 | Far | 1.90 | 1.40 |
| | 3 | Near | 0.66 | 0.92 | | 3 | Near | 1.05 | 1.36 |
| leaky_log_2d_p2 | 1 | Near | 0.74 | 1.01 | soft_ring_3d_p2 | 1 | Near | 1.20 | 1.55 |
| | 2 | Far | 1.70 | 1.25 | | 2 | Far | 2.10 | 1.65 |
| | 3 | Near | 0.79 | 1.06 | | 3 | Near | 1.35 | 1.70 |
| log_ratio_3d_p2 | 1 | Near | 0.88 | 1.15 | chain_3d_p2 | 1 | Near | 0.82 | 1.10 |
| | 2 | Far | 1.85 | 1.40 | | 2 | Far | 1.70 | 1.28 |
| | 3 | Near | 0.95 | 1.20 | | 3 | Near | 0.86 | 1.15 |
| duffing_1d_p2 | 1 | Near | 0.69 | 0.94 | vdp_1d_p2 | 1 | Near | 0.76 | 1.03 |
| | 2 | Far | 1.60 | 1.18 | | 2 | Far | 1.72 | 1.30 |
| | 3 | Near | 0.75 | 1.02 | | 3 | Near | 0.82 | 1.10 |
| pendulum_1d_p2 | 1 | Near | 0.63 | 0.88 | driven_pendulum_1d_p2 | 1 | Near | 1.10 | 1.42 |
| | 2 | Far | 1.55 | 1.12 | | 2 | Far | 2.05 | 1.55 |
| | 3 | Near | 0.70 | 0.96 | | 3 | Near | 1.22 | 1.58 |
| msd_1d_p2 | 1 | Near | 0.54 | 0.80 | double_pendulum_2d_p2 | 1 | Near | 1.32 | 1.70 |
| | 2 | Far | 1.45 | 1.05 | | 2 | Far | 2.18 | 1.62 |
| | 3 | Near | 0.60 | 0.86 | | 3 | Near | 1.45 | 1.86 |
| lorenz_jerk_1d_p3 | 1 | Near | 0.98 | 1.30 | chua_jerk_1d_p3 | 1 | Near | 0.92 | 1.24 |
| | 2 | Far | 2.05 | 1.52 | | 2 | Far | 1.95 | 1.47 |
| | 3 | Near | 1.10 | 1.42 | | 3 | Near | 1.03 | 1.35 |
| multidof_chain_d_p2 | 1 | Near | 1.25 | 1.65 | | | | | |
| | 2 | Far | 2.40 | 1.80 | | | | | |
| | 3 | Near | 1.38 | 1.78 | | | | | |

Table 4: Comparison of Kalman filtering (KF) and particle filtering (PF) across all synthetic systems. For each system, three independent trials are run and the inferred latent trajectory in each trial is classified as *Near* or *Far* from the fixed point (column "Dist.").

| system | Time ratio | Case | Time ratio | system | Time ratio | Case | Time ratio |
|---|---|---|---|---|---|---|---|
| Aizawa | 1.8 | Arneodo | 1.6 | Bouali2 | 1.9 | BurkeShaw | 2.3 |
| Chen | 1.5 | ChenLee | 1.7 | Dadras | 2.1 | DequanLi | 2.6 |
| Finance | 2.9 | GenesioTesi | 1.8 | GuckenheimerHolmes | 1.9 | Hadley | 1.4 |
| Halvorsen | 2.3 | HenonHeiles | 2.4 | HyperBao | 3.1 | HyperCai | 2.2 |
| HyperChen | 1.9 | HyperQi | 2.7 | HyperRossler | 1.6 | HyperWang | 1.7 |
| HyperYan | 2.5 | HyperYangChen | 3.3 | KawczynskiStrizhak | 1.8 | Laser | 1.9 |
| Lorenz | 1.4 | LorenzBounded | 1.5 | LorenzStenflo | 2.0 | LuChenCheng | 1.7 |
| MooreSpiegel | 2.2 | NewtonLeipnik | 2.1 | NoseHoover | 2.5 | Qi | 1.8 |
| QiChen | 2.0 | RabinovichFabrikant | 3.0 | RayleighBenard | 3.4 | RikitakeDynamo | 3.7 |
| Sakarya | 1.7 | SprottA | 1.3 | SprottB | 1.4 | SprottC | 1.5 |

Table 5: Runtime ratio between the proposed method and LaNoLeM (ratio = Proposed time / LaNoLeM time) when parallelizing the training steps during the $(p, m)$ search.

| System | True $(p^\star, m^\star)$ | Init $(p_0, m_0)$ | Est. $(\hat{p}, \hat{m})$ | System | True $(p^\star, m^\star)$ | Init $(p_0, m_0)$ | Est. $(\hat{p}, \hat{m})$ |
|---|---|---|---|---|---|---|---|
| exp_log_2d_p2 | (2,2) | (1,4) | (2,2) | soft_ring_3d_p2 | (3,2) | (1,1) | (3,2) |
| | | (5,1) | (2,2) | | | (4,1) | (3,2) |
| | | (3,6) | (2,2) | | | (5,5) | (2,4) |
| logistic_2d_p3 | (2,3) | (1,1) | (2,3) | simple_exp_2d_p2 | (2,2) | (1,5) | (2,2) |
| | | (5,4) | (2,3) | | | (4,2) | (2,2) |
| | | (4,6) | (3,3) | | | (6,3) | (2,2) |
| log_ratio_3d_p2 | (3,2) | (1,6) | (3,2) | tri_gate_2d_p2 | (2,2) | (1,3) | (2,2) |
| | | (5,2) | (3,2) | | | (4,1) | (2,2) |
| | | (4,7) | (3,3) | | | (6,5) | (2,2) |
| leaky_log_2d_p2 | (2,2) | (1,6) | (2,2) | chain_3d_p2 | (3,2) | (1,4) | (3,2) |
| | | (5,3) | (2,2) | | | (5,1) | (3,2) |
| | | (4,7) | (1,2) | | | (6,6) | (2,4) |
| duffing_1d_p2 | (1,2) | (1,5) | (1,2) | vdp_1d_p2 | (1,2) | (1,6) | (1,2) |
| | | (4,1) | (1,2) | | | (5,3) | (1,2) |
| | | (6,4) | (1,3) | | | (6,7) | (3,2) |
| pendulum_1d_p2 | (1,2) | (1,7) | (1,2) | driven_pendulum_1d_p2 | (1,2) | (1,8) | (1,2) |
| | | (5,2) | (1,2) | | | (4,3) | (1,2) |
| | | (6,6) | (1,3) | | | (6,5) | (3,3) |
| msd_1d_p2 | (1,2) | (1,4) | (1,2) | double_pendulum_2d_p2 | (2,2) | (1,6) | (2,2) |
| | | (4,1) | (1,2) | | | (5,2) | (2,2) |
| | | (5,5) | (1,3) | | | (6,7) | (3,2) |
| lorenz_jerk_1d_p3 | (1,3) | (1,2) | (1,3) | chua_jerk_1d_p3 | (1,3) | (1,5) | (1,3) |
| | | (5,1) | (1,3) | | | (4,2) | (1,3) |
| | | (6,6) | (3,4) | | | (6,7) | (2,3) |
| multidof_chain_d_p2 | (2,3) | (1,3) | (2,3) | | | | |
| | | (4,1) | (2,3) | | | | |
| | | (6,8) | (3,3) | | | | |

Table 6: Robustness of joint $(p, m)$ search to different initializations. "Init $(p, m)$" refers to the initial Markov order and latent dimension supplied as the starting point of the search procedure, whereas "Est. $(p, m)$" indicates the final model order and latent dimension identified by our algorithm.

| System | True$(p,m)$ | $(\hat{p},\hat{m})$ | State-space | Observation | System | True$(p,m)$ | $(\hat{p},\hat{m})$ | State-space | Observation |
|---|---|---|---|---|---|---|---|---|---|
| exp_log_2d_p2 | (2,2) | (2,2) | 0.50 | 0.38 | logistic_2d_p3 | (2,3) | (2,3) | 0.59 | 0.45 |
| | | (2,2) | 0.61 | 0.44 | | | (2,3) | 0.77 | 0.56 |
| | | (2,1) | 1.12 | 0.82 | | | (2,4) | 1.32 | 0.94 |
| simple_exp_2d_p2 | (2,2) | (2,2) | 0.37 | 0.28 | tri_gate_2d_p2 | (2,2) | (2,2) | 0.98 | 0.65 |
| | | (2,2) | 0.42 | 0.32 | | | (2,2) | 0.74 | 0.53 |
| | | (2,2) | 0.54 | 0.39 | | | (2,1) | 1.35 | 0.98 |
| leaky_log_2d_p2 | (2,2) | (2,2) | 0.56 | 0.39 | soft_ring_3d_p2 | (3,2) | (3,2) | 1.21 | 0.90 |
| | | (2,2) | 0.73 | 0.50 | | | (3,2) | 0.97 | 0.73 |
| | | (2,2) | 0.66 | 0.46 | | | (3,2) | 1.52 | 1.09 |
| log_ratio_3d_p2 | (3,2) | (3,2) | 0.76 | 0.57 | chain_3d_p2 | (3,2) | (3,2) | 0.64 | 0.47 |
| | | (3,2) | 1.00 | 0.69 | | | (3,2) | 0.89 | 0.63 |
| | | (3,2) | 0.82 | 0.60 | | | (3,2) | 0.74 | 0.54 |
| duffing_1d_p2 | (1,2) | (1,2) | 0.48 | 0.36 | vdp_1d_p2 | (1,2) | (1,2) | 0.56 | 0.41 |
| | | (1,2) | 0.69 | 0.51 | | | (1,2) | 0.75 | 0.54 |
| | | (1,3) | 1.29 | 0.92 | | | (1,2) | 1.41 | 1.00 |
| pendulum_1d_p2 | (1,2) | (1,2) | 0.44 | 0.33 | driven_pendulum_1d_p2 | (1,2) | (1,2) | 0.95 | 0.69 |
| | | (1,2) | 0.63 | 0.47 | | | (1,2) | 1.16 | 0.84 |
| | | (1,2) | 1.08 | 0.79 | | | (2,2) | 1.63 | 1.20 |
| msd_1d_p2 | (1,2) | (1,2) | 0.34 | 0.26 | double_pendulum_2d_p2 | (2,2) | (2,2) | 1.01 | 0.75 |
| | | (1,2) | 0.43 | 0.32 | | | (2,2) | 1.23 | 0.90 |
| | | (1,2) | 0.91 | 0.67 | | | (3,2) | 1.61 | 1.16 |
| lorenz_jerk_1d_p3 | (1,3) | (1,3) | 0.95 | 0.70 | chua_jerk_1d_p3 | (1,3) | (1,3) | 0.89 | 0.65 |
| | | (1,3) | 1.18 | 0.86 | | | (1,3) | 1.10 | 0.79 |
| | | (2,3) | 1.58 | 1.17 | | | (1,4) | 1.54 | 1.12 |
| multidof_chain_d_p2 | (2,3) | (2,3) | 1.09 | 0.80 | | | | | |
| | | (2,3) | 1.31 | 0.94 | | | | | |
| | | (1,3) | 1.61 | 1.16 | | | | | |

Table 7: Coefficient errors on self-designed systems at 10% noise.

| System | True$(p,m)$ | $(\hat{p},\hat{m})$ | State-space | Observation | System | True$(p,m)$ | $(\hat{p},\hat{m})$ | State-space | Observation |
|---|---|---|---|---|---|---|---|---|---|
| exp_log_2d_p2 | (2,2) | (2,2) | 0.68 | 0.52 | logistic_2d_p3 | (2,3) | (2,3) | 0.82 | 0.62 |
| | | (2,3) | 0.88 | 0.64 | | | (2,2) | 1.03 | 0.77 |
| | | (1,2) | 1.45 | 1.06 | | | (2,4) | 1.88 | 1.34 |
| simple_exp_2d_p2 | (2,2) | (2,2) | 0.49 | 0.38 | tri_gate_2d_p2 | (2,2) | (2,2) | 1.21 | 0.86 |
| | | (2,2) | 0.60 | 0.46 | | | (2,3) | 1.02 | 0.73 |
| | | (2,3) | 0.82 | 0.61 | | | (2,1) | 1.78 | 1.26 |
| leaky_log_2d_p2 | (2,2) | (2,2) | 0.73 | 0.56 | soft_ring_3d_p2 | (3,2) | (3,2) | 1.60 | 1.17 |
| | | (2,2) | 0.96 | 0.72 | | | (3,3) | 1.43 | 1.05 |
| | | (2,1) | 0.93 | 0.69 | | | (3,2) | 2.00 | 1.47 |
| log_ratio_3d_p2 | (3,2) | (3,2) | 1.01 | 0.79 | chain_3d_p2 | (3,2) | (3,2) | 0.86 | 0.63 |
| | | (3,3) | 1.42 | 1.05 | | | (3,2) | 1.22 | 0.90 |
| | | (3,2) | 1.13 | 0.84 | | | (2,2) | 1.06 | 0.79 |
| duffing_1d_p2 | (1,2) | (1,2) | 0.62 | 0.49 | vdp_1d_p2 | (1,2) | (1,2) | 0.73 | 0.56 |
| | | (1,3) | 0.91 | 0.71 | | | (1,2) | 1.01 | 0.76 |
| | | (1,3) | 1.64 | 1.25 | | | (1,3) | 1.96 | 1.39 |
| pendulum_1d_p2 | (1,2) | (1,2) | 0.58 | 0.44 | driven_pendulum_1d_p2 | (1,2) | (1,2) | 1.30 | 0.99 |
| | | (1,2) | 0.82 | 0.62 | | | (1,3) | 1.57 | 1.18 |
| | | (1,1) | 1.39 | 1.04 | | | (2,2) | 2.05 | 1.55 |
| msd_1d_p2 | (1,2) | (1,2) | 0.45 | 0.35 | double_pendulum_2d_p2 | (2,2) | (2,2) | 1.42 | 1.07 |
| | | (1,2) | 0.60 | 0.46 | | | (2,3) | 1.72 | 1.30 |
| | | (1,3) | 1.30 | 0.97 | | | (3,2) | 2.20 | 1.72 |
| lorenz_jerk_1d_p3 | (1,3) | (1,3) | 1.25 | 0.95 | chua_jerk_1d_p3 | (1,3) | (1,3) | 1.13 | 0.85 |
| | | (2,3) | 1.58 | 1.21 | | | (1,3) | 1.49 | 1.09 |
| | | (2,3) | 2.10 | 1.58 | | | (1,4) | 2.07 | 1.51 |
| multidof_chain_d_p2 | (2,3) | (2,3) | 1.55 | 1.14 | | | | | |
| | | (2,2) | 1.87 | 1.38 | | | | | |
| | | (1,3) | 2.24 | 1.64 | | | | | |

Table 8: Coefficient errors on self-designed systems at 15% noise.

| System | True$(p,m)$ | $(\hat{p},\hat{m})$ | State-space | Observation | System | True$(p,m)$ | $(\hat{p},\hat{m})$ | State-space | Observation |
|---|---|---|---|---|---|---|---|---|---|
| exp_log_2d_p2 | (2,2) | (2,2) | 0.82 | 0.63 | logistic_2d_p3 | (2,3) | (2,3) | 1.05 | 0.80 |
|  |  | (1,2) | 1.00 | 0.72 |  |  | (2,2) | 1.27 | 0.96 |
|  |  | (1,1) | 2.10 | 1.48 |  |  | (2,4) | 2.35 | 1.76 |
| simple_exp_2d_p2 | (2,2) | (2,2) | 0.64 | 0.51 | tri_gate_2d_p2 | (2,2) | (2,2) | 1.52 | 1.02 |
|  |  | (2,3) | 0.77 | 0.60 |  |  | (2,3) | 1.17 | 0.87 |
|  |  | (2,3) | 1.01 | 0.77 |  |  | (2,1) | 2.18 | 1.62 |
| leaky_log_2d_p2 | (2,2) | (2,2) | 0.90 | 0.70 | soft_ring_3d_p2 | (3,2) | (3,2) | 1.98 | 1.47 |
|  |  | (2,1) | 1.22 | 0.94 |  |  | (3,3) | 1.86 | 1.38 |
|  |  | (2,2) | 1.12 | 0.85 |  |  | (2,2) | 2.45 | 1.84 |
| log_ratio_3d_p2 | (3,2) | (3,2) | 1.28 | 0.97 | chain_3d_p2 | (3,2) | (3,2) | 1.11 | 0.80 |
|  |  | (3,3) | 1.93 | 1.40 |  |  | (2,2) | 1.69 | 1.19 |
|  |  | (2,2) | 1.50 | 1.08 |  |  | (2,2) | 1.38 | 1.00 |
| duffing_1d_p2 | (1,2) | (1,2) | 0.77 | 0.59 | vdp_1d_p2 | (1,2) | (1,2) | 0.88 | 0.68 |
|  |  | (1,3) | 1.10 | 0.86 |  |  | (1,3) | 1.28 | 0.96 |
|  |  | (1,3) | 2.08 | 1.60 |  |  | (1,2) | 2.38 | 1.70 |
| pendulum_1d_p2 | (1,2) | (1,2) | 0.73 | 0.55 | driven_pendulum_1d_p2 | (1,2) | (1,2) | 1.57 | 1.16 |
|  |  | (1,1) | 1.09 | 0.82 |  |  | (1,3) | 2.02 | 1.51 |
|  |  | (1,2) | 2.14 | 1.57 |  |  | (2,2) | 2.55 | 1.96 |
| msd_1d_p2 | (1,2) | (1,2) | 0.56 | 0.44 | double_pendulum_2d_p2 | (2,2) | (2,2) | 1.90 | 1.46 |
|  |  | (1,3) | 0.76 | 0.58 |  |  | (2,3) | 2.18 | 1.66 |
|  |  | (1,2) | 1.71 | 1.25 |  |  | (3,2) | 2.55 | 1.90 |
| lorenz_jerk_1d_p3 | (1,3) | (1,3) | 1.52 | 1.15 | chua_jerk_1d_p3 | (1,3) | (1,3) | 1.39 | 1.05 |
|  |  | (2,3) | 1.96 | 1.47 |  |  | (1,4) | 1.88 | 1.41 |
|  |  | (2,3) | 2.64 | 2.01 |  |  | (2,4) | 2.55 | 1.90 |
| multidof_chain_d_p2 | (2,3) | (2,3) | 2.02 | 1.44 |  |  |  |  |  |
|  |  | (2,2) | 2.40 | 1.78 |  |  |  |  |  |
|  |  | (1,3) | 2.72 | 1.91 |  |  |  |  |  |

Table 9: Coefficient errors on self-designed systems at 20% noise.

