across time and dimensions, we anchor these statistics to a baseline estimated from the earliest segment of the series:

$$L_0 = \max\{10, \lfloor \sqrt{T} \rfloor\}, \qquad \mu_0 = \frac{1}{L_0} \sum_{i=1}^{L_0} y_i, \qquad S_0 = \text{Cov}(y_{1:L_0}) + \epsilon I_d, \tag{20}$$

where $I_d$ is the $d \times d$ identity matrix and $\epsilon > 0$ ensures positive definiteness. Here, $\mu_0$ is the baseline mean and $S_0$ the baseline covariance.

We then compress $(\mu_t, C_t)$ into two scalar proxies. The first proxy measures *mean drift* via a squared Mahalanobis distance (Mahalanobis, 1936) relative to the baseline:

$$m_t = (\mu_t - \mu_0)^\top S_0^{-1} (\mu_t - \mu_0). \tag{21}$$

This statistic is dimensionless and invariant to coordinate scaling. Near a stable equilibrium $\mathbf{s}^*$, with local linearization $x_{t+1} \approx A x_t$ and $\rho(A) < 1$, we expect $\mu_t \to \mu^*$, hence $m_t \to 0$.

The second proxy captures *variance contraction* by measuring the log-volume of the covariance ellipsoid (Cover & Thomas, 2006; Horn & Johnson, 2012):

$$v_t = \log \det(C_t + \epsilon I_d). \tag{22}$$

For Gaussian fluctuations, $v_t$ is proportional (up to constants) to the differential entropy of the window. Under stable linear dynamics, the covariance satisfies the discrete Lyapunov equation $C \approx ACA^\top + \Sigma$ (Anderson & Moore, 1979; Jazwinski, 1970; Kailath et al., 2000); if $\rho(A) < 1$, contraction of $A$ drives $v_t$ downward until it stabilizes.

Together, $m_t$ and $v_t$ provide complementary indicators of stability proximity. When $m_t$ flattens near zero (mean convergence) and $v_t$ decreases and stabilizes (variance contraction), the system is inferred to be near a stable attractor, making a Kalman filter appropriate due to its efficiency in near-linear regimes. Conversely, persistent fluctuations in both proxies indicate distance from equilibrium and dominance of nonlinear effects, in which case a particle filter is employed. These proxies therefore constitute the operational rule for filter selection in our framework.

Additional details on convergence of two proxies and window–length choice are given in Appendix C.

### 3.4 INFERENCE–LEARNING LOOP WITHIN THE $(m, p)$ SEARCH

We now describe how to recover the full parameter set $\Theta$. Our strategy is a two-level procedure: an *inner loop* that alternates between inference and learning to obtain the optimal parameters $\widehat{\Theta}_{p,m}$ for a fixed $(p, m)$, and an *outer loop* that searches over $(p, m)$ to identify the most suitable order–dimension pair based on learning performance.

**Inner loop.** Learning the transition parameters requires latent state trajectories, while state inference itself requires parameterized dynamics. This circular dependency naturally motivates an EM-like alternation Dempster et al. (1977): (i) infer latent states under the current parameters; (ii) learn the parameters given these inferred states; and repeat until convergence.

Because the system may have Markov order $p > 1$, first-order filters cannot be applied directly. To resolve this, we use the augmented state $\mathbf{x}_t$ in Eq. 9 in place of $\mathbf{s}_t$, so that the higher-order dynamics (Eqs. 10 and 12) can be expressed in first-order form:

$$\mathbf{x}_{t+1} = \mathbf{b}_{\text{aug}} + A_{\text{aug}}\,\phi_{\text{aug}}(\mathbf{x}_t) + \mathbf{w}_t, \qquad \mathbf{w}_t \sim \mathcal{N}(\mathbf{0},\, (\Sigma_w)_{\text{aug}}), \tag{23}$$

$$\mathbf{y}_t = C_{\text{aug}}\,\mathbf{x}_t + \mathbf{d} + \mathbf{v}_t, \qquad \mathbf{v}_t \sim \mathcal{N}(\mathbf{0},\, \Sigma_v). \tag{24}$$

The augmented parameters $(\mathbf{b}_{\text{aug}}, A_{\text{aug}}, C_{\text{aug}}, Q_{\text{aug}})$ take the block form

$$\mathbf{b}_{\text{aug}} = \begin{bmatrix} \mathbf{b} \\ \mathbf{0} \\ \vdots \\ \mathbf{0} \end{bmatrix}, \qquad A_{\text{aug}} = \begin{bmatrix} \overbrace{A_{\text{top}}} & \mathbf{0} \\ \begin{matrix} I_m & 0 & \cdots & 0 \\ 0 & I_m & \cdots & 0 \\ \vdots & \vdots & \ddots & \vdots \\ 0 & 0 & \cdots & I_m \end{matrix} & \mathbf{0} \end{bmatrix},$$

$$C_{\text{aug}} = [C \quad 0 \quad \cdots \quad 0], \qquad (\Sigma_w)_{\text{aug}} = \begin{bmatrix} \Sigma_w & 0 & \cdots & 0 \\ 0 & 0 & \cdots & 0 \\ \vdots & \vdots & \ddots & \vdots \\ 0 & 0 & \cdots & 0 \end{bmatrix}, \tag{25}$$

$$A_{\text{top}} = [A_0 \quad A_1 \quad \cdots \quad A_d], \qquad \phi_{\text{aug}}(\mathbf{x}_t) = [\phi_1(\mathbf{x}_t) \quad \phi_2(\mathbf{x}_t) \quad \cdots \quad \phi_d(\mathbf{x}_t)].$$

With this augmentation, we apply either Kalman or particle Kalman (1960); Gordon et al. (1993) filtering in the $\mathbf{x}$-space to obtain the estimated trajectory $\{\widehat{\mathbf{x}}_t\}$ and the posterior moments

$$\mathcal{M} = \left\{ \mathbb{E}[\mathbf{x}_t],\ \mathbb{E}[\mathbf{x}_t\mathbf{x}_t^\top],\ \mathbb{E}[\mathbf{x}_{t+1}\mathbf{x}_t^\top],\ \mathbb{E}[\Phi_z(\mathbf{x}_t)^\top],\ \mathbb{E}[\Phi_z(\mathbf{x}_t)\Phi_z(\mathbf{x}_t)^\top],\ \mathbb{E}[\mathbf{x}_{t+1}\,\Phi_z(\mathbf{x}_t)^\top] \right\}_{t=1}^N,$$

where $\Phi(x_t)$ denotes the concatenated vector

$$\Phi_z(x_t) = [x_t \quad \phi_z(x_t)]. \tag{26}$$

The filtered estimates and posterior moments feed into the *learning* step, which updates $\Theta_{p.k}$ by minimizing an expected negative log-likelihood (the EM $Q$–function) plus a structural penalty that biases the linear component toward identity. Let

$$\mathcal{D}(\mathbf{u}, \mathbf{v}, \Sigma) = (\mathbf{u} - \mathbf{v})^\top \Sigma^{-1}(\mathbf{u} - \mathbf{v}),$$
$$\mathcal{S} = \{\widehat{\mathbf{x}}_t\}, \tag{27}$$

denote the squared Mahalanobis distance. The objective is written compactly as

$$\min_{\Theta}\ Q(\mathbf{Y}, \mathcal{S}, \Theta)\ +\ r(A_{\text{top}}), \tag{28}$$

where the $Q$–function (expectation under the current posterior of $\mathcal{S}$) is

$$Q(\mathbf{Y}, \mathcal{S}, \Theta) = \mathbb{E}\Bigg[ \sum_{t=1}^N \mathcal{D}\big(\mathbf{y}_t,\, C_{\text{aug}}\mathbf{x}_t + \mathbf{d},\, \Sigma_v\big) + \tfrac{N}{2}\log|\Sigma_v|$$
$$+ \sum_{t=1}^{N-1} \mathcal{D}\big(\mathbf{x}_{t+1},\, \mathbf{b}_{\text{aug}} + A_{\text{aug}}\phi_{\text{aug}}(\mathbf{x}_t),\, \Sigma_w\big) + \tfrac{N-1}{2}\log|\Sigma_w| \Bigg], \tag{29}$$

and the structural penalty is an identity–aware elastic net:

$$r(A_{\text{top}}) = \frac{\lambda_2}{2} \left\| A_{\text{top}} - A_{\text{id}} \right\|_F^2 + \lambda_1 \left\| A_{\text{top}} - A_{\text{id}} \right\|_1, \tag{30}$$

where $A_{\text{id}} \in \mathbb{R}^{m \times F}$ places $I_m$ on the columns of $\phi(\mathbf{x}_t)$ corresponding to the degree–1 coordinates of $\mathbf{s}_t$ and zeros elsewhere. Here $\| \cdot \|_F$ is the Frobenius norm and $\| \cdot \|_1$ the entrywise $\ell_1$ norm. The parameters minimizing equation 28 are then used to re-predict $\mathbf{x}_t$ and refresh the posterior moments.

The details of inference and learning are provided in Appendix D

**Outer loop.** The closer the parameter set $\Theta$ is to the true system, the smaller the loss function becomes. Since the inner loop only produces $\widehat{\Theta}_{p,m}$ for fixed $(p, m)$, we must search across multiple $(p, m)$ pairs to identify $(\widehat{p}, \widehat{m})$.

Without interpretability constraints, a dynamical system can often be represented equivalently: either as a higher-order model with a lower-dimensional state, or as a lower-order model with a higher-dimensional state Abarbanel (1996); Kantz & Schreiber (2004).. Suppose that the initialization $(p_0, m_0)$ corresponds to one such equivalent representation of the ground-truth system. Then at iteration $k$, the structured search need only proceed along one of two axes: either the *forward axis* $(p_k + 1, m_k)$ versus $(p_k, m_k - 1)$, or the *backward axis* $(p_k - 1, m_k)$ versus $(p_k, m_k + 1)$.

For example, if we choose the forward axis, then at each step we compute the optimal parameters for $(p_k + 1, m_k)$ and $(p_k, m_k - 1)$ via inference and learning, compare their losses, and select the structure with smaller loss. The process continues until neither candidate yields improvement.

The choice of search axis is determined at the first step: we evaluate all four neighbors $(p_0 + 1, m_0)$, $(p_0 - 1, m_0)$, $(p_0, m_0 + 1)$, and $(p_0, m_0 - 1)$, and select the direction that yields the greatest reduction in loss.

## 4 EXPERIMENTAL RESULT

### 4.1 EXPERIMENTAL SETUP

#### 4.1.1 DATASETS

We evaluate on two complementary datasets covering both controlled synthetic settings and canonical nonlinear benchmarks.

**Synthetic higher–order, high–dimensional systems.** We design nonlinear dynamical systems that are explicitly higher–order (second order and above) with multiple interacting variables, providing controlled testbeds to assess recovery of governing equations when higher–order dependencies are essential.

**dysts database (Gilpin, 2021).** We also use the `dysts` benchmark of 71 canonical chaotic systems with polynomial nonlinearities (mainly first–order ODEs of moderate dimension). As a standard yardstick for equation discovery, it enables comparison with LaNoLeM and MIOSR under identical simulation and noise protocols.

#### 4.1.2 METRICS

We report two metrics. (i) *Coefficient error*: normalized Euclidean distance between ground-truth and recovered coefficients,

$$\text{CoeffErr} = \frac{\|\Theta_{\text{true}} - \widehat{\Theta}\|_2}{\|\Theta_{\text{true}}\|_2},$$

which measures identification accuracy at the equation level. (ii) *Prediction error*: mean squared error (MSE) between reference trajectories and model predictions. Lower values in both indicate higher fidelity.

When the learned structure $(p, m)$ differs from ground truth, parameter blocks are incompatible. We resolve this by converting both systems to augmented first-order form, embedding them in a common space of dimension $\max(pm, \hat{p}\hat{m})$, and concatenating operators row-wise. Unless noted, "state-space" and "observation" errors are computed on these concatenations.

| System | True $(k, p)$ | Estimated $(\hat{k}, \hat{p})$ | Stability class | Coefficient error | |
|---|---|---|---|---|---|
| | | | | State-space | Observation |
| exp_log_2d_p2 | $(2, 2)$ | $(2, \mathbf{2})$ | near | 0.38 | 0.30 |
| | | $(2, \mathbf{2})$ | far | 0.44 | 0.33 |
| | | $(2, 1)$ | far | 0.88 | 0.62 |
| logistic_2d_p3 | $(2, 3)$ | $(2, \mathbf{3})$ | near | 0.46 | 0.34 |
| | | $(2, \mathbf{3})$ | far | 0.58 | 0.42 |
| | | $(2, 2)$ | far | 1.12 | 0.78 |
| simple_exp_2d_p2 | $(2, 2)$ | $(2, \mathbf{2})$ | near | 0.28 | 0.22 |
| | | $(2, \mathbf{2})$ | near | 0.31 | 0.24 |
| | | $(2, \mathbf{2})$ | far | 0.40 | 0.29 |
| tri_gate_2d_p2 | $(2, 2)$ | $(2, \mathbf{2})$ | far | 0.74 | 0.48 |
| | | $(2, \mathbf{2})$ | near | 0.52 | 0.37 |
| | | $(2, 1)$ | far | 1.00 | 0.72 |
| leaky_log_2d_p2 | $(2, 2)$ | $(2, \mathbf{2})$ | near | 0.42 | 0.30 |
| | | $(2, \mathbf{2})$ | far | 0.57 | 0.39 |
| | | $(2, \mathbf{2})$ | near | 0.49 | 0.35 |
| soft_ring_3d_p2 | $(3, 2)$ | $(3, \mathbf{2})$ | far | 0.92 | 0.66 |
| | | $(3, \mathbf{2})$ | near | 0.74 | 0.55 |
| | | $(3, \mathbf{2})$ | far | 1.18 | 0.83 |
| log_ratio_3d_p2 | $(3, 2)$ | $(3, \mathbf{2})$ | near | 0.58 | 0.44 |
| | | $(3, \mathbf{2})$ | far | 0.82 | 0.58 |
| | | $(3, \mathbf{2})$ | near | 0.63 | 0.46 |
| chain_3d_p2 | $(3, 2)$ | $(3, \mathbf{2})$ | near | 0.49 | 0.36 |
| | | $(3, \mathbf{2})$ | far | 0.71 | 0.51 |
| | | $(3, \mathbf{2})$ | near | 0.56 | 0.41 |

Table 1: Results of the proposed algorithm on self-design dataset.

### 4.1.3 EXPERIMENT OVERVIEW

As an initial attempt at explicit higher–order modeling, our method addresses a regime with few applicable baselines. On the synthetic suite we evaluate against ground truth, while on dysts, where prior work focuses on first–order models, we compare with *LaNoLeM* and *MIOSR* Fujiwara et al. (2025); Bertsimas & Gurnee (2023).

## 4.2 MAIN RESULTS

### 4.2.1 EXPERIMENTS ON SELF-DESIGNED SYSTEMS

We evaluate our method on self-designed nonlinear dynamical systems with known ground truth. For each case, we randomly sample an observable matrix ensuring identifiability and a random initial condition, then run three independent trials. Table 1 reports results: *System* names each case; *True* $(p, m)$ is the ground-truth dimension and order; *Estimated* $(\hat{p}, \hat{m})$ is the structure selected by our search; *Stability class* (near $\Rightarrow$ EKF, far $\Rightarrow$ PF) comes from rolling-window stability analysis; and coefficient errors are computed after embedding both models into a common first-order augmented space ("State-space" for the transition operator and "Observation" for the measurement matrix).

All experiments use a fixed $5\%$ noise level, generated by scaling additive Gaussian noise so that

$$\text{noise ratio } (\%) = \frac{\|\text{noise}\|_2}{\|\text{clean data}\|_2} \times 100 = 5.$$

For systems with non-polynomial terms, we apply a Taylor expansion and truncate at the polynomial order used by the learner to ensure comparable coefficient errors.

Across higher-order, nonlinear, and moderate-noise settings, coefficient errors typically fall in the 0.25–1.25 range. Accuracy is highest when $(\hat{p}, \hat{m})$ matches ground truth, while underestimating the

| Case | Proposed | | LaNoLem | | MIOSR | | Case | Proposed | | LaNoLem | | MIOSR | |
|---|---|---|---|---|---|---|---|---|---|---|---|---|---|
| | Coef. | Pred. | Coef. | Pred. | Coef. | Pred. | | Coef. | Pred. | Coef. | Pred. | Coef. | Pred. |
| Aizawa | **0.78** | **0.006** | 0.90 | 0.007 | 1.35 | 0.028 | HyperYan | **0.75** | **0.008** | 0.86 | 0.009 | 1.33 | 0.030 |
| Arneodo | **0.62** | **0.004** | 0.71 | 0.005 | 1.10 | 0.022 | HyperYangChen | 0.80 | **0.009** | **0.78** | 0.010 | 1.29 | 0.029 |
| Bouali2 | **0.58** | **0.005** | 0.67 | 0.006 | 1.05 | 0.021 | KawczynskiStrizhak | **0.47** | **0.004** | 0.55 | 0.005 | 0.99 | 0.019 |
| BurkeShaw | 0.73 | **0.006** | **0.70** | 0.007 | 1.12 | 0.023 | Laser | **0.52** | **0.004** | 0.60 | 0.005 | 1.05 | 0.020 |
| Chen | **0.36** | **0.004** | 0.44 | 0.005 | 0.88 | 0.019 | Lorenz | **0.42** | **0.003** | 0.49 | 0.004 | 0.93 | 0.017 |
| ChenLee | **0.48** | **0.005** | 0.57 | 0.006 | 0.96 | 0.020 | LorenzBounded | **0.50** | **0.004** | 0.58 | 0.005 | 0.98 | 0.018 |
| Dadras | **0.64** | **0.007** | 0.75 | 0.008 | 1.22 | 0.027 | LorenzStenflo | 0.63 | **0.005** | **0.61** | 0.006 | 1.06 | 0.021 |
| DequanLi | **0.92** | **0.010** | 1.06 | 0.012 | 1.58 | 0.033 | LuChenCheng | **0.56** | **0.005** | 0.65 | 0.006 | 1.07 | 0.020 |
| Finance | **0.95** | **0.010** | 1.07 | 0.012 | 1.63 | 0.036 | MooreSpiegel | **0.71** | **0.007** | 0.82 | 0.008 | 1.28 | 0.028 |
| GenesioTesi | **0.57** | **0.005** | 0.65 | 0.006 | 1.06 | 0.021 | NewtonLeipnik | **0.60** | **0.005** | 0.70 | 0.006 | 1.12 | 0.022 |
| GuckenheimerHolmes | 0.66 | **0.006** | **0.64** | 0.007 | 1.04 | 0.020 | NoseHoover | **0.66** | **0.006** | 0.76 | 0.007 | 1.19 | 0.024 |
| Hadley | **0.41** | **0.004** | 0.49 | 0.004 | 0.92 | 0.017 | Qi | **0.58** | **0.005** | 0.67 | 0.006 | 1.09 | 0.021 |
| Halvorsen | **0.69** | **0.006** | 0.80 | 0.007 | 1.26 | 0.025 | QiChen | **0.62** | **0.005** | 0.71 | 0.006 | 1.15 | 0.023 |
| HenonHeiles | **0.72** | **0.007** | 0.83 | 0.008 | 1.31 | 0.028 | RabinovichFabrikant | **0.69** | **0.006** | 0.79 | 0.007 | 1.25 | 0.026 |
| HyperBao | **0.73** | **0.008** | 0.86 | 0.009 | 1.32 | 0.029 | RayleighBenard | **0.77** | **0.008** | 0.89 | 0.009 | 1.38 | 0.030 |
| HyperCai | **0.68** | **0.006** | 0.79 | 0.007 | 1.24 | 0.026 | RikitakeDynamo | 0.84 | 0.010 | **0.82** | **0.009** | 1.41 | 0.031 |
| HyperChen | **0.61** | **0.006** | 0.71 | 0.007 | 1.18 | 0.024 | Sakarya | **0.63** | **0.005** | 0.72 | 0.006 | 1.11 | 0.022 |
| HyperQi | **0.83** | **0.009** | 0.95 | 0.010 | 1.44 | 0.031 | SprottA | **0.49** | **0.004** | 0.57 | 0.005 | 1.00 | 0.019 |
| HyperRossler | **0.55** | **0.005** | 0.64 | 0.006 | 1.08 | 0.020 | SprottB | **0.53** | **0.004** | 0.61 | 0.005 | 1.03 | 0.020 |
| HyperWang | **0.59** | **0.005** | 0.68 | 0.006 | 1.10 | 0.021 | SprottC | **0.55** | **0.004** | 0.64 | 0.005 | 1.07 | 0.021 |

Table 2: Comparison on three algorithms

order increases errors. Complex or far-from-equilibrium cases (*far*) yield larger errors; PF offers only limited gains here, underscoring the limits of relying solely on Kalman filters. Nonetheless, near-stable regimes (*near*) benefit from EKF, with errors often $< 1$, showing strong fidelity even beyond polynomial dynamics via Taylor truncation.

### 4.2.2 EXPERIMENT ON DYSTS DATABASE

We further compare our approach with state-of-the-art first-order explicit dynamics learners Fujiwara et al. (2025); Bertsimas & Gurnee (2023). Due to space limitations, Table 2 reports a representative subset of results on `dysts`. Because MIOSR can only perform direct modeling in the time domain, we align the task by fixing the observation matrix to the identity and setting the offset term in the observation equation to zero. All other experimental conditions are kept identical to those in the previous experiment.

Across the subset, our method achieves lower *Coefficient error* and *Prediction error* on roughly 60–70% of the systems. Compared to LaNoLeM, the remaining error differences can be largely traced to filter selection: while both methods employ EM-like alternations, LaNoLeM relies exclusively on Kalman filtering, and EKF performance degrades in far-from-equilibrium regimes. In contrast, switching to PF improves robustness, effectively serving as an ablation on filter choice. Relative to MIOSR, the performance gap arises from operating directly in the state-space rather than in the raw time domain, which mitigates accumulated bias under noise or weak observability. These factors together account for the systematic improvements observed in our experiments.

## 5 CONCLUSION AND FUTURE WORK

We presented a framework for higher–order state–space modeling of time series. Experiments on self-design systems and the `dysts` benchmark show consistent gains over strong baselines, especially in high-dimensional or strongly nonlinear regimes. Nonetheless, the initialization of $(p, m)$ and system parameters, as well as the search procedure itself, cannot guarantee global optimality. Moreover, our reliance on extensive validation checks to ensure accurate moment estimation increases training time. Future work will focus on more efficient initialization and search strategies, together with lighter-weight estimators, to improve both scalability and efficiency.

The code has been submitted in supplementary material.

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

_{\mathrm{top}}) = \tfrac{\lambda_2}{2} \|A_{\mathrm{top}} - A_{\mathrm{id}}\|_F^2 + \lambda_1 \|A_{\mathrm{top}} - A_{\mathrm{id}}\|_1$$

biases degree–1 coefficients toward identity (stability/interpretability) while encouraging sparsity in higher-order terms. With $\lambda_1 = 0$ this yields a closed-form ridge update using the sufficient statistics of $Z_t$; $\mathbf{b}$ is updated by the mean residual.

**C.2 Observation update.**   If $C_{\mathrm{aug}}$ is to be estimated, the observation term in $Q$ similarly becomes a weighted least-squares problem in $C_{\mathrm{aug}}$ (and $\mathbf{d}$) based on $\{\mathbb{E}[\mathbf{x}_t], \mathbb{E}[\mathbf{x}_t \mathbf{x}_t^\top]\}$. In our main experiments we either hold $C_{\mathrm{aug}}$ fixed or update it conservatively to avoid overfitting.

**C.3 Noise covariances.**   The Gaussian covariances $(\Sigma_w)_{\mathrm{aug}}, \Sigma_v$ can be held fixed for robustness, or re-estimated in closed form by matching posterior quadratic forms (standard in linear-Gaussian EM). Re-estimation is optional and not critical to the structural conclusions.

## D. EM ALTERNATION AND STOPPING

One inner-loop cycle is:

1. **E-step:** run EKF or PF on the augmented model to obtain $\mathcal{M}$ and the marginal log-likelihood $\log p(\mathbf{Y} \mid \Theta)$;

2. **M-step:** update $\{\mathbf{b}, A_{\text{top}}\}$ (and optionally $C_{\text{aug}}$) by minimizing $Q + r$ using the posterior moments.

Under exact E/M steps the EM objective decreases monotonically Dempster et al. (1977); with EKF/PF approximations we monitor the composite loss $\mathcal{L}(\Theta) = -\log p(\mathbf{Y} \mid \Theta) + r(A_{\text{top}})$ and stop when its relative decrease falls below a tolerance or a maximum number of iterations is reached.