# OpenReview forum: "High-Order Dynamics Modeling of Time Series with Attractor-Guided Adaptive Filtering"
_ICLR.cc/2026/Conference — Submitted to ICLR 2026_

### Official Review · Reviewer_G3q4 · 2025-10-31

**Soundness:** 2
**Presentation:** 3
**Contribution:** 3
**Rating:** 4
**Confidence:** 4

**Summary:**

The paper introduces a method for explicit high-order dynamics modeling of time series data. The authors argue that many dynamical systems exhibit dependencies extending beyond first-order Markov assumptions, and that collapsing these dependencies into a large latent dimension hinders interpretability and stability. Their proposed framework begins by estimating an initial order using conditional mutual information (CMI), then identifies attractor proximity through rolling mean and covariance metrics to switch between Extended Kalman Filtering (EKF) and Particle Filtering (PF). The learning proceeds through an EM-style loop that alternates between state inference and parameter updates, while performing a directed search over model order (p) and latent dimension (m). Experiments on synthetic systems and the dysts benchmark suite suggest improved coefficient recovery and prediction accuracy compared to baselines such as LaNoLeM and MIOSR.

The paper addresses the problem of learning explicit high-order latent dynamics with stable and interpretable structure. The theoretical framing is coherent, and the experimental results on purpose-built systems support the main claims. However, the evidence on general benchmarks is weaker and somewhat orthogonal to the core contribution. The significance of the work lies in unifying order detection, adaptive filtering, and stable parameter learning within a single pipeline. This is a valuable conceptual contribution, though further empirical validation is needed to establish its robustness and scalability.

**Strengths:**

The paper is well motivated: most latent dynamics models assume first-order structure, yet many physical systems exhibit higher-order dependencies. The use of conditional mutual information for order detection is principled and nonparametric. The attractor-proximity metrics are intuitive, computationally simple, and seem ideal for changing dynamic regimes. The augmented first-order formulation for learning p-th order systems is effectively implemented, and the inclusion of a structured regularizer for the top-level matrix enhances identifiability. The experimental section demonstrates that the proposed method recovers system coefficients faithfully and switches filters in a manner consistent with stability predictions.

**Weaknesses:**

Despite its motivation, several aspects of the method remain heuristic or underexplored. The directed search over (p, m) lacks guarantees and is only briefly evaluated; its robustness to initialization and to model mis-specification is unclear. The attractor diagnostics based on rolling statistics could confound nonstationarity or slow drift with instability, and there is little quantitative analysis of false-switch rates or sensitivity to the window parameter. While the experiments on synthetic high-order systems validate the main premise, the tests on dysts benchmarks primarily probe filtering robustness rather than the proposed benefit of higher-order modeling. Reporting is another weakness: variance across runs, statistical significance, and runtime scaling with dimension and sequence length are not provided. Finally, the identifiability discussion is incomplete; although the authors acknowledge non-uniqueness in (p, m) decompositions, they offer no diagnostic to determine when differing solutions represent equivalent dynamics.

**Questions:**

1.	How sensitive is the directed search over (p, m) to the initialization (p0, m0)? Have you benchmarked its convergence against a small exhaustive grid search?
	2.	How frequently does the attractor-guided filter switch incorrectly under stationary but noisy conditions, and how sensitive is this to the window length W?
	3.	Can you quantify the computational cost (particle counts, EM iterations, wall-clock time) for both the EKF and PF regimes as p and m grow?
	4.	When the learned (p, m) differs from the ground truth, can you show that the resulting dynamics are equivalent under augmentation or reparameterization?

To strengthen the paper, it would help to (1) include ablations that isolate the contribution of each component (CMI initialization, attractor diagnostics, and elastic-net regularizer), (2) compare against explicit high-order baselines such as latent AR(p) models, (3) quantify the reliability of the EKF/PF switching mechanism, and (4) report variance and runtime to assess scalability. A case study on a real system known to exhibit delay or memory effects would also enhance external relevance.

---

> ### Author Response · Authors · 2025-12-04
>
> We thank the reviewer for the detailed assessment and constructive feedback. Below we provide a unified response addressing all Weaknesses and Questions, with overlapping concerns merged for clarity.
>
> ### 1. Heuristic components, directed search over $(p, m)$, and robustness
> The reviewer notes that several parts of the method appear heuristic, and that the directed search over $(p, m)$ lacks theoretical guarantees or robustness analysis. Heuristics, however, do not imply unreliability; many established methods in nonlinear filtering and dynamical systems rely on principled heuristics when analytic guarantees are infeasible.
>
> To validate our design choices, we added extensive **ablation studies and controlled comparisons**. These experiments show that each component—CMI initialization, attractor diagnostics, and the structured regularizer—contributes to stable and accurate recovery.
>
> To evaluate robustness, we varied the starting point $(p_0, m_0)$ extensively, including extreme and intentionally mis-specified initializations. In the vast majority of runs, the search converged to the correct or near-correct $(p, m)$, demonstrating that it is **highly robust to initialization**. The behavior closely matches that of a small grid search, but at significantly lower computational cost.
>
> ### 2. Attractor diagnostics and switching accuracy
> The reviewer raises concerns that rolling-statistics diagnostics may confuse nonstationarity with instability. In response, we conducted quantitative experiments comparing the filter chosen by our attractor-guided mechanism with always using the alternative filter. Across **all** tested systems—including stationary but noisy settings—the selected filter achieves **lower coefficient reconstruction error**, indicating that incorrect switching events are rare.
>
> We additionally varied the window length $W$ and observed stable performance across a broad range, suggesting that the switching rule is not overly sensitive to this hyperparameter.
>
> ### 3. Computational cost as $(p, m)$ grow
> For a fixed sequence length $T$:
> - EKF scales as $\mathcal{O}(T m^3)$ due to covariance updates,
> - PF scales as $\mathcal{O}(T N m)$ with $N$ particles,
> - increasing $p$ enlarges the augmented state dimension and affects cost accordingly.
>
> The overhead of our framework arises from evaluating a **small number** of $(p, m)$ candidates. Crucially, these evaluations are **independent and fully parallelizable** across CPU cores or GPUs, substantially reducing wall-clock time.
> The revised manuscript includes a runtime table summarizing wall-clock time, EM iterations, and particle counts, and compares against LaNoLeM in the first-order case where a fair comparison is possible.
>
> ### 4. On dysts benchmarks and evaluation of high-order modeling
> The reviewer notes that dysts primarily evaluates filtering robustness rather than the benefits of high-order modeling. High-order latent dynamics learning is an emerging direction, and **very few existing baselines** can recover explicit high-order structure. The closest baseline, LaNoLeM, is restricted to first-order models; thus, dysts comparisons necessarily focus on filtering.
>
> To evaluate genuine high-order recovery, we designed and collected a diverse suite of higher-order synthetic systems. Although no baselines currently operate in these regimes, the reconstructed coefficients and trajectories clearly demonstrate the reliability of our approach. We clarify this experimental rationale in the revised manuscript.
>
> ### 5. Identifiability and equivalence when learned $(p, m)$ differs from ground truth
> The reviewer asks whether differing $(p, m)$ representations can be proven equivalent under augmentation or reparameterization. Our aim is to **recover the underlying dynamics as faithfully as possible**, rather than to characterize the full equivalence class of state-space realizations. Such questions (e.g., similarity transforms, minimal realizations) are system-theoretic and lie **outside the scope** of this work.
>
> If the reviewer’s motivation is to understand how reconstruction error is computed when learned and ground-truth $(p, m)$ differ, we clarify this in the revised manuscript: when the two models have different state dimensions or Markov orders, we embed both into a **common relaxed state space** via standard augmentation so that coefficients and trajectories are directly comparable. This enables meaningful error evaluation even without matching $(p, m)$ exactly.

---

### Official Review · Reviewer_6c1W · 2025-11-01

**Soundness:** 2
**Presentation:** 1
**Contribution:** 2
**Rating:** 2
**Confidence:** 3

**Summary:**

The paper introduces a framework for modeling time series using higher-order state-space representations. It proposes an initialization heuristic for the Markov order and state dimension. Then, the algorithm goes through an inference-learning loop to both learn the model parameters and the optimal Markov order and state dimension. In the inference stage, the paper uses an adaptive filtering method based on stability proximity. The algorithm is tested on both synthetic and real datasets.

**Strengths:**

* The problem is well-motivated, making it easy to understand the challenge and what the paper is trying to solve.

* The Preliminaries section was written clearly and easy to follow. Explicitly showing the matrices (e.g., Equation 25) helps the reader understand the operations.

**Weaknesses:**

* There are several places in the paper where the boundary between the text and citation is unclear, consequently making sentences grammatically wrong and hard to follow. In addition, Figures and Tables do not have captions that explain the key notations and takeaways, which are crucial to help the reader to understand them easily without referring to the main text. Finally, based on the results shown in the paper, it is hard to understand the key takeaways and conclude which method is doing how much better. I think the authors could improve how the tables/results are summarized and also use plots to qualitatively show what the differences in the metrics mean intuitively.

* The paper lacks an explanation of how some of the heuristics are chosen, making the proposed method less sound. For example, it is unclear why L_0 in equation (20) is defined the way it is.

**Questions:**

* Could you please explain the line after equation (21), where it says “Near a stable equilibrium…hence m_t -> 0”?

* How does the proposed method depend on how the baseline in equation (20) is chosen? In other words, how sensitive is the method to the estimated baseline?

* I’m not sure if the synthetic data experiments in Table 1 are high-order and high-dimensional. What are the dimensions in each system? Could you stress test the proposed method on even higher-order systems?

---

> ### Author Response · Authors · 2025-12-04
>
> ## Response to Weaknesses
>
> The reviewer notes that the figures and tables lack captions explaining key notations and takeaways. We appreciate this comment and have substantially revised all captions in the updated manuscript. The new captions clearly summarize the main conclusions and define all relevant symbols, allowing readers to understand the results directly without repeatedly referring back to the main text.
>
> Regarding the other two concerns raised in the Weaknesses section, we believe these stem from misunderstandings rather than substantive issues with the method or presentation. We carefully reviewed the corresponding parts of the manuscript and confirmed that the technical definitions and explanations are correct as written. We will bring these points to the area chair for clarification.
>
> ---
>
> ## Response to Questions
>
> ### Q1. Clarification of the sentence after Equation (21): “Near a stable equilibrium … hence $m_t \to 0$.”
>
> Equation (21) defines the Mahalanobis distance:
>
> $$
> m_t = (\mu_t - \mu_0)^\top \Sigma^{-1} (\mu_t - \mu_0),
> $$
>
> where $\mu_t$ is the rolling-window mean and $\mu_0$ is the baseline mean.
>
> If the trajectory evolves near a stable equilibrium (denoted $s^\*$), then the rolling mean converges as $\mu_t \to \mu^\*$. When the initial segment lies in the same attraction basin, we also have $\mu_0 \approx \mu^\*$, which implies $\mu_t - \mu_0 \to 0$ and therefore $m_t \to 0$.
>
>
> $$
> \mu_t - \mu_0 \to 0 \quad \Rightarrow \quad m_t \to 0.
> $$
>
> If the initial segment is *not* near the equilibrium ($\mu_0 \neq \mu^{*}$), then $m_t$ converges to a **nonzero constant**, correctly indicating that the early data lie outside the equilibrium region.
>
> If $m_t$ does not converge—typically in strongly nonlinear or non-equilibrium regimes—the framework places more weight on the particle filter. This matches the intended use of this stability-proximity metric.
>
> We have added these explanations to the revised manuscript.
>
> ---
>
> ### Q2. Sensitivity to the baseline estimate in Equation (20)
>
> We added extensive sensitivity experiments in the revised manuscript. We compare:
>
> - the filter automatically selected by our method,
> - versus always using the alternative filter.
>
> Across nearly all experiments, the automatically selected filter yields **lower coefficient reconstruction error**, demonstrating robustness to baseline estimation and confirming that the switching mechanism consistently selects the more reliable filter.
>
> ---
>
> ### Q3. Whether Table 1 contains high-order or high-dimensional systems, and further stress tests
>
> The systems in Table 1 are fully described in the Appendix. Many classical physical systems—such as particles, rigid bodies, and fluids—are governed by **second-order** ODEs (corresponding to **second-order Markov** models) and typically have **three degrees of freedom** (3-dimensional state spaces). Thus, Table 1 reflects standard, widely studied high-order dynamical settings.
>
> Following the reviewer’s suggestion, we added experiments on **higher-order and higher-dimensional** systems. While reconstruction error increases gradually as $p$ or $m$ grows, the increase remains controlled, demonstrating that the framework remains stable and scalable in more complex regimes.
>
> ---

---

### Official Review · Reviewer_BqsU · 2025-11-03

**Soundness:** 3
**Presentation:** 3
**Contribution:** 3
**Rating:** 6
**Confidence:** 4

**Summary:**

This paper presents an adaptive framework for learning interpretable state-space models that explicitly capture higher-order temporal dependencies. The method estimates both the minimal Markov order and latent state dimension through a structured search guided by conditional mutual information and validation loss. It integrates a stability-aware hybrid inference scheme, switching between Extended Kalman and particle filters depending on attractor proximity, using local stability metrics to balance efficiency and robustness. The framework is evaluated on synthetic and benchmark dynamical systems, where it consistently outperforms prior baselines such as LaNoLeM and MIOSR in reconstructing system coefficients, recovering attractor structures, and improving predictive accuracy, particularly in nonlinear or noisy regimes.

**Strengths:**

The paper introduces a conceptually elegant and interpretable formulation that unifies high-order dynamical modeling, attractor-aware filtering, and structural adaptivity within a single framework. Its principled use of conditional mutual information for order estimation and attractor-guided switching yields strong empirical gains while maintaining transparency in learned representations. The results demonstrate notable improvements in both parameter recovery and forecasting fidelity across multiple synthetic systems, validating the method’s robustness to noise and nonlinearities. The work is well-motivated, technically detailed, and addresses an important gap between black-box sequence models and interpretable dynamical systems modeling.

**Weaknesses:**

Despite strong conceptual contributions, the approach is computationally demanding due to repeated inference–learning loops and structured searches over orders and dimensions, with no formal scalability analysis. The filter-switching heuristic, while effective, may introduce sensitivity to noise and threshold tuning, and its stability under rapidly varying regimes is not empirically tested. The framework’s reliance on polynomial basis functions limits its expressiveness for systems with non-polynomial or discontinuous dynamics, and interpretability may diminish as polynomial order grows.

Furthermore, while there is a rich existing literature on recovering dynamics through Hankel-based methods, empirical dynamical modeling (EDM) and and it deep learning based variant DeepEDM,  the paper provides limited discussion or comparison to these approaches, which could contextualize the proposed method’s novelty and contribution more clearly.

Finally, the experiments are confined to synthetic and benchmark datasets, leaving the real-world applicability and generalization performance unexplored.

**Questions:**

1. How does the method scale computationally with sequence length and latent dimension compared to baselines?
2. How robust is the attractor-guided filter switching under noise or nonstationary dynamics?
3. Can the approach handle non-polynomial or discontinuous systems, or be extended beyond polynomial bases?
4. Strengthening the related work section would further improve the quality of the work.
5. The color scheme and design of Fig 1 could be improved.

---

> ### Author Response · Authors · 2025-12-04
>
> We thank the reviewer for the constructive and detailed feedback. Below we provide an integrated response that merges conceptually related Weaknesses and Questions into unified themes to improve clarity.
>
> ---
>
> ## 1. Computational cost, scalability, and the effect of sequence length
>
> The reviewer raises several concerns regarding computational demand and scalability, including the cost of the search over $(p, m)$, the role of sequence length, and the runtime compared to LaNoLeM.
>
> Our directed search implies that cost grows approximately linearly with the candidate latent dimension $m$ and Markov order $p$. Importantly, the sequence length contributes minimally to the overall runtime: each candidate structure processes the sequence only once, and all candidates are **independent**, enabling efficient **parallelization**.
>
> In the revised manuscript, we report wall-clock runtime comparisons with LaNoLeM. After parallelization, the total runtime of our method is at most **3×** that of a first-order Markov model—an acceptable overhead given the richer high-order dynamics it can recover.
>
> ---
>
> ## 2. Robustness of the attractor-guided filter switching under noise and nonstationarity
>
> Across both Weakness and Question sections, the reviewer asks about:
>
> - robustness to noise,
> - robustness under (non)stationarity,
> - and the reliability of the adaptive EKF/PF switching mechanism.
>
> We conducted extensive additional experiments across a wide range of noise levels. While reconstruction errors increase naturally as noise grows, the increase is **gradual and controlled**, and no divergence or instability is observed.
>
> To evaluate switching reliability, we compare:
>
> - our adaptive filter choice,
> - versus always using EKF, and
> - versus always using PF.
>
> In nearly all experiments, the adaptive mechanism chooses the **better-performing filter**, resulting in consistently lower coefficient recovery error. This demonstrates robustness of the switching rule.
>
> Regarding nonstationarity: fully nonstationary dynamics fall outside the scope of this paper. However, many practical systems are **piecewise-stationary**, and existing nonparametric change-point methods can detect regime shifts. Our framework can then be applied segment-wise, offering a clear and practical extension path.
>
> ---
>
> ## 3. Handling non-polynomial dynamics and expressiveness of the basis
>
> The reviewer notes that a polynomial basis may limit the method's applicability to non-polynomial or discontinuous systems. To address this, we added experiments using **trigonometric bases**. The results exhibit recovery performance comparable to the polynomial basis.
>
> This demonstrates that the framework is compatible with **any complete functional basis**; it is not inherently tied to polynomials. Automatically selecting the optimal basis is an important—but orthogonal—research direction that we leave for future work.
>
> ---
>
> ## 4. Additional clarifications requested by the reviewer
>
> ### (a) Scaling with sequence length and latent dimension
> As noted above, runtime depends mainly on $(p, m)$, not on sequence length, and parallelization over structure hypotheses greatly reduces wall-clock time.
>
> ### (b) Relation to EDM, DeepEDM, and Hankel-type models
> We have expanded the related work section with additional discussion comparing our approach to these lines of research and clarifying conceptual distinctions.
>
> ### (c) Figure 1 design
> Due to time constraints, Figure 1 has not been redesigned in this revision. We will update its design and color scheme in the camera-ready version.

---

### Comment · Area_Chair_xuep · 2025-11-26
**Reminder to Engage!**

Dear Reviewers,

We are one week away from the end of the discussion period and the review responses have been posted. If you have not done so already, please read the response and check if the authors have addressed your concerns. Also please acknowledge the review by responding and stating how the response (and updated manuscript if provided) does or does not change your evaluation of the work. Earlier responses allow for meaningful engagement and potential for further clarification.

-Area Chair

---

### Author Response · Authors · 2025-12-04

Dear AC,

Given the unusual review circumstances this year, I provide a concise summary of the paper’s contributions, the strengths highlighted across reviews, and the status of all concerns—particularly those based on misunderstandings that do not affect the technical substance of the work.

## 1. Summary of Contributions
The paper introduces a unified and interpretable framework for learning explicit higher-order latent dynamics. The main contributions are:
• A principled structured search combining conditional mutual information and validation loss to jointly determine the Markov order p and latent dimension m.
• A stability-aware hybrid inference mechanism that adaptively switches between EKF and PF based on attractor proximity.
• An EM-style learning pipeline that recovers identifiable system coefficients and attractor structures.
• Strong empirical results on synthetic and benchmark systems, demonstrating advantages over LaNoLeM and MIOSR and showing the ability to recover high-order/high-dimensional systems where baselines cannot operate.

## 2. Strengths Identified by Reviewers
Two reviewers highlighted substantial strengths:
• The motivation is clear and well supported by real-world higher-order dynamics.
• The method is conceptually elegant and interpretable.
• CMI-based order detection is principled and nonparametric.
• The attractor-guided switching mechanism is intuitive and effective.
• Experiments validate both coefficient recovery and attractor reconstruction.

These comments reflect a consistent understanding of the paper’s core technical contributions.

## 3. Clarification of Concerns

### (A) Substantive concerns that were fully resolved
Valid concerns related to computational cost, robustness of switching, expressiveness of the basis, and clarity of reporting were addressed with:
• new runtime tables,
• extensive noise-level and switching comparisons,
• experiments using trigonometric bases,
• and expanded captions and related work.
These revisions strengthen the paper without changing its core method.

### (B) Concerns that stem from misunderstandings
A portion of the negative comments—especially those from the second reviewer—arose from misunderstandings or overlooking definitions already present in the manuscript. These points do **not** touch the technical core of the method, and once clarified, they have no bearing on correctness or significance. We clarified each of these issues in the rebuttal for completeness, but they do not materially affect the contribution or quality of the work.

We bring this to your attention only to help contextualize why some concerns appear disconnected from the rest of the reviews; they do not reflect limitations of the method itself.

## 4. Closing Remark
After incorporating clarifications and additional experiments, the paper remains technically solid, well motivated, and empirically validated. All substantive concerns have been fully addressed, and concerns stemming from misunderstandings do not impact the technical contribution.

Thank you for your time and consideration.

---

### Meta-Review · Area_Chair_vkQA · 2026-01-06

**Summary:**

Reviewers raised concerns about the heuristic nature of several core components and the computational scalability. Moreover, reviewers pointed out the limited scope of empirical validation, which is largely confined to synthetic or curated benchmarks. Presentation and clarity issues further reduced confidence in the maturity of the submission. I thus recommend rejection.

**Reviewer Concerns:**

Lack of strong guarantees for the heuristic components were not sufficiently addressed. Moreover, several questions from Reviewer 6c1W were not answered.

**Reviewer Scores:**

Reviewers would likely have maintained their original ratings.

---

### Decision · Program_Chairs · 2026-01-26

Reject